# Six domesticated PiggyBac transposases together carry out programmed DNA elimination in *Paramecium*

**Julien Bischerour[1]\*, Simran Bhullar[2,3], Cyril Denby Wilkes[1], Vinciane Régnier[1,4], Nathalie Mathy[1†], Emeline Dubois[1], Aditi Singh[2], Estienne Swart[2‡], Olivier Arnaiz[1], Linda Sperling[1], Mariusz Nowacki[2], Mireille Bétermier[1]\***

[1]Institute for Integrative Biology of the Cell (I2BC), CEA, CNRS, Univ. Paris-Sud, Université Paris-Saclay, Gif-sur-Yvette, France; [2]Institute of Cell Biology, University of Bern, Bern, Switzerland; [3]Institut de Biologie de l'Ecole Normale Supérieure, Paris, France; [4]Univ Paris Diderot, Paris, France

**Abstract** The domestication of transposable elements has repeatedly occurred during evolution and domesticated transposases have often been implicated in programmed genome rearrangements, as remarkably illustrated in ciliates. In *Paramecium*, PiggyMac (Pgm), a domesticated PiggyBac transposase, carries out developmentally programmed DNA elimination, including the precise excision of tens of thousands of gene-interrupting germline Internal Eliminated Sequences (IESs). Here, we report the discovery of five groups of distant Pgm-like proteins (PgmLs), all able to interact with Pgm and essential for its nuclear localization and IES excision genome-wide. Unlike Pgm, PgmLs lack a conserved catalytic site, suggesting that they rather have an architectural function within a multi-component excision complex embedding Pgm. PgmL depletion can increase erroneous targeting of residual Pgm-mediated DNA cleavage, indicating that PgmLs contribute to accurately position the complex on IES ends. DNA rearrangements in *Paramecium* constitute a rare example of a biological process jointly managed by six distinct domesticated transposases.

DOI: https://doi.org/10.7554/eLife.37927.001

\*For correspondence:
julien.bischerour@i2bc.paris-saclay.fr (JB);
mireille.betermier@i2bc.paris-saclay.fr (MB)

Present address: †RDP, ENS-Lyon, Lyon, France; ‡Max Planck Institute of Developmental Biology, Tuebingen, Germany

Competing interests: The authors declare that no competing interests exist.

## Introduction

The mobility of DNA transposons is ensured by their self-encoded transposase (reviewed in *Hickman and Dyda, 2015*). The most commonly studied transposases harbor an RNase H-related catalytic domain including three conserved acidic residues DD(D/E) and have been grouped into distinct superfamilies (*Curcio and Derbyshire, 2003*; *Wicker et al., 2007*; *Hickman et al., 2010*). During evolution, exaptation of transposon-borne genes has sometimes given rise to novel cellular functions through a process called domestication (*Volff, 2006*; *Jangam et al., 2017*). Several instances of domesticated DD(D/E) transposases have been reported, some of which still exhibit at least partial catalytic activity. The *Transib*-originating Rag1 protein catalyzes V(D)J recombination of vertebrate immunoglobulin genes (*Kapitonov and Jurka, 2005*; *Huang et al., 2016*); SETMAR, a partially active domesticated *mariner* transposase, is involved in DNA double-strand break repair in primates (*Liu et al., 2007*; *Kim et al., 2014*); α3, domesticated from a *hAT* transposon, and Kat1, domesticated from a *Mutator*-like element, carry out mating type switching in the yeast *Kluyveromyces lactis* (*Barsoum et al., 2010*; *Rajaei et al., 2014*). CENP-B, related to *mariner* elements, serves as a centromere-binding factor, but its ancestral catalytic domain is no longer required for its function (*Mateo and González, 2014*).

Transposases from the *piggyBac* family have repeatedly been domesticated in eukaryotes (*Bouallègue et al., 2017*). In mammals, five *PGBD* (*piggyBac*-derived) genes have been identified, but their cellular function has so far remained elusive (*Sarkar et al., 2003*). The most ancient, *PGBD5* (*Pavelitz et al., 2013*), encodes a protein with a highly divergent catalytic domain that is active for DNA cleavage and transposition (*Henssen et al., 2015*) and promotes DNA rearrangements in human cancers (*Henssen et al., 2017*). *PGBD1* and *2* are conserved in mammals, but their encoded proteins have lost the DDD catalytic triad characteristic of active PiggyBac (PB) transposases and their cellular function is unknown. *PGBD3* and *4* are restricted to primates. Pgbd3, expressed as a fusion with the Cockayne Syndrome CSB transcription factor, does not carry an intact catalytic site, but has retained specific DNA binding activity to *piggyBac*-related genomic sequences, which may expand the gene network that is transcriptionally regulated by CSB-Pgbd3 (*Gray et al., 2012*; *Weiner and Gray, 2013*). In contrast, Pgbd4 harbors a conserved DDD triad, but its cellular function is unknown. Remarkably, catalytically active domesticated PB transposases play an essential role during developmentally programmed genome rearrangements in the ciliates *Paramecium* and *Tetrahymena* (*Baudry et al., 2009*; *Cheng et al., 2010*; *Vogt and Mochizuki, 2013*; *Cheng et al., 2016*; *Dubois et al., 2017*).

Ciliates are unicellular eukaryotes characterized by their nuclear dimorphism, with two types of nuclei coexisting in the same cytoplasm (*Prescott, 1994*). The diploid germline micronucleus (MIC), transcriptionally inactive during vegetative growth, undergoes meiosis and transmits the parental genetic information to the zygotic nucleus during sexual reproduction. The highly polyploid somatic macronucleus (MAC), streamlined for gene expression and essential for cell growth, is fragmented and destroyed at each sexual cycle and a new MAC develops from a mitotic copy of the zygotic nucleus. During MAC development, massive genome amplification takes place and, following a few endoduplication rounds,~30% of germline sequences are removed from the somatic genome in *P. tetraurelia* (*Arnaiz et al., 2012*) and *T. thermophila* (*Hamilton et al., 2016*). In both species, DNA elimination requires the introduction of programmed DNA double-strand breaks (DSB) at the boundaries of eliminated sequences (*Saveliev and Cox, 1996*; *Gratias and Bétermier, 2003*). Two modes of sexual reproduction have been described in *Paramecium*: conjugation and autogamy, a self-fertilization process (reviewed in *Betermier and Duharcourt, 2014*). In both processes, programmed DNA elimination targets two types of germline sequences. Repeated sequences, for example transposable elements (TEs) or minisatellites, are removed in association with chromosome fragmentation. In addition, the precise excision of 45,000 single-copy non-coding Internal Eliminated Sequences (IESs), which interrupt 47% of all genes in the germline genome, allows proper assembly of functional open reading frames in the somatic genome, essential for the survival of sexual progeny. *Paramecium* IESs are short (93% shorter than 150 bp), non-coding sequences, whose size follows a sinusoid-shaped distribution with a periodicity equal to the helical pitch of double-stranded B DNA (*Arnaiz et al., 2012*). IESs are flanked with a conserved TA dinucleotide at each end; a single TA remains at the excision site. IES ends define a loosely conserved 8 bp consensus sequence (5'-TAYAGYNR-3'), of unclear mechanistic significance. Indeed, how the excision machinery accurately targets IES ends remains an open question.

IES excision is a precise 'cut-and-close' mechanism that starts with the introduction of DNA DSBs centered on the flanking TAs (*Gratias and Bétermier, 2003*). PiggyMac (Pgm), a domesticated PB transposase with an intact DDD catalytic motif, is responsible for DNA cleavage (*Baudry et al., 2009*; *Dubois et al., 2017*) and the resulting DSBs are repaired through the classical non-homologous end joining pathway (C-NHEJ) (*Kapusta et al., 2011*; *Allen et al., 2017*). Tight coupling of DSB introduction and repair is thought to be ensured by the assembly of a Pgm/Ku complex required for DNA cleavage (*Marmignon et al., 2014*). Here, we report the discovery of five groups of paralogous *Paramecium* domesticated PB transposases, designated as Pgm-like(s) (PgmL), that appear to be novel essential components of the Pgm-associated complex. Using a combination of RNAi-mediated knockdowns (KDs), immunofluorescence microscopy and whole genome sequencing, we show that each PgmL group is essential for Pgm nuclear localization during the sexual cycle and efficient genome-wide IES excision. In some KDs, residual Pgm complexes lacking one PgmL partner are still detected in the developing MAC and retain partial activity. However, they tend to incorrectly target IES excision boundaries, resulting in excision errors. Our data, as a whole, indicate that six groups of domesticated PB transposases, including one catalytically active subunit (Pgm) and five additional partners (PgmL), act together to carry out IES excision. We discuss a model, in which

PgmLs associate with Pgm, favor and stabilize its nuclear localization and ensure the precise positioning of DNA cleavage.

## Results

### Novel domesticated PB transposase genes in the *P. tetraurelia* genome

Two structural domains can be predicted in Pgm using the Pfam protein family database (*Finn et al., 2016*) (http://pfam.xfam.org/, *Figure 1A*). The first domain (PF13843 or DDE_Tnp1_7) encompasses the RNase H fold-related catalytic domain found in DD(D/E) transposases. The second domain (DDE_Tnp_1-like zinc-ribbon) corresponds to a cysteine-rich domain (CRD), essential for Pgm activity (*Dubois et al., 2017*). Using a Hidden Markov Model (HMM) search, we discovered that nine putative Pgm-related proteins, hereafter designated as PiggyMac-like (PgmL) proteins, are encoded by the *P. tetraurelia* somatic genome (*Supplementary file 2*). A Pfam domain search predicted that the DDE_Tnp1_7 transposase domain is conserved in all PgmLs (*Figure 1A*). The DDE_Tnp_1-like zinc-ribbon domain was not systematically found using this approach, but alignment of protein sequences confirmed that all PgmLs carry a CRD (*Figure 1A*, *Figure 1—figure supplement 1* and *Supplementary file 3*).

PgmL-encoding genes form five groups of paralogs. *PGML1* and *PGML2* each are single genes, whereas *PGML3* is composed of three genes: *PGML3a and b* are duplicates from the most recent whole genome duplication (WGD) that took place during evolution of the *P. aurelia* group of species (*Aury et al., 2006*), *PGML3c* arose from an earlier 'intermediate' WGD. Similarly, *PGML4a and b* on the one hand and *PGML5a and b* on the other are paralogs from the most recent WGD. Genes from distinct *PGML* groups do not share nucleotide sequence homology with each other and their encoded proteins are very divergent in sequence and domain organization (*Figure 1A* and *Figure 1—figure supplement 2*). Within each group, however, WGD paralogs encode highly similar proteins. Analysis of published genome assemblies confirmed the conservation of at least one representative of each *PGML* group in other *P. aurelia* species (*McGrath et al., 2014b*) (*Supplementary file 2*, *Figure 1—figure supplement 2*). Evidence was also obtained that all *PGML* groups are present in a more distant species, *P. caudatum* (*McGrath et al., 2014a*) (*Supplementary file 2*).

The predicted PgmL proteins have different lengths, ranging from 578 to 1085 residues (*Figure 1A*). The domain organization of PgmL1 and PgmL2 is close to that of canonical PB transposases, whereas other PgmLs carry additional domains: PgmL3, PgmL4 and PgmL5 have a carboxy-terminal extension predicted to be rich in coiled-coil; an amino-terminal extension with no homology to any known structure is found in PgmL4 and PgmL5. While the presence and position of a 'DDD' motif of three aspartic acids in the conserved RNase H domain is essential for the catalytic activity of PB transposases (*Mitra et al., 2008*), we did not find a complete DDD triad in any PgmL using a combination of sequence alignment and secondary structure prediction (*Figure 1B*, *Supplementary file 1* and *4*). PgmL1, PgmL2, PgmL3 and PgmL5 do not carry any conserved D residue, while only two out of three are found in PgmL4 at the expected first and third positions of the triad (D619/617 and D785/783). Given that a single mutation in the catalytic triad is sufficient to completely abolish in vitro activity of the PB transposase from *Trichoplusia ni* (*Mitra et al., 2008*), it is unlikely that PgmLs are still catalytically active.

### *PGMLs* are expressed during autogamy and localize in the new developing MAC

We analyzed previously published high-throughput sequencing data obtained from polyadenylated RNAs extracted during three standard autogamy time-courses of *P. tetraurelia* (*Arnaiz et al., 2017*), and found that *PGMLs* are all specifically induced and co-expressed with *PGM* during new MAC development, when programmed genome rearrangements take place (*Figure 2A*). We observe maximal expression levels of all genes around 5 to 11 hr (T5 to T11) following the T0 time-point that corresponds to the stage when 50% of cells in the population have fragmented their old MAC.

The development-specific expression of *PGMLs* suggests that their encoded proteins may be implicated in DNA rearrangements during MAC development. To confirm protein production and follow the cellular localization of PgmLs during autogamy, we raised specific antibodies against

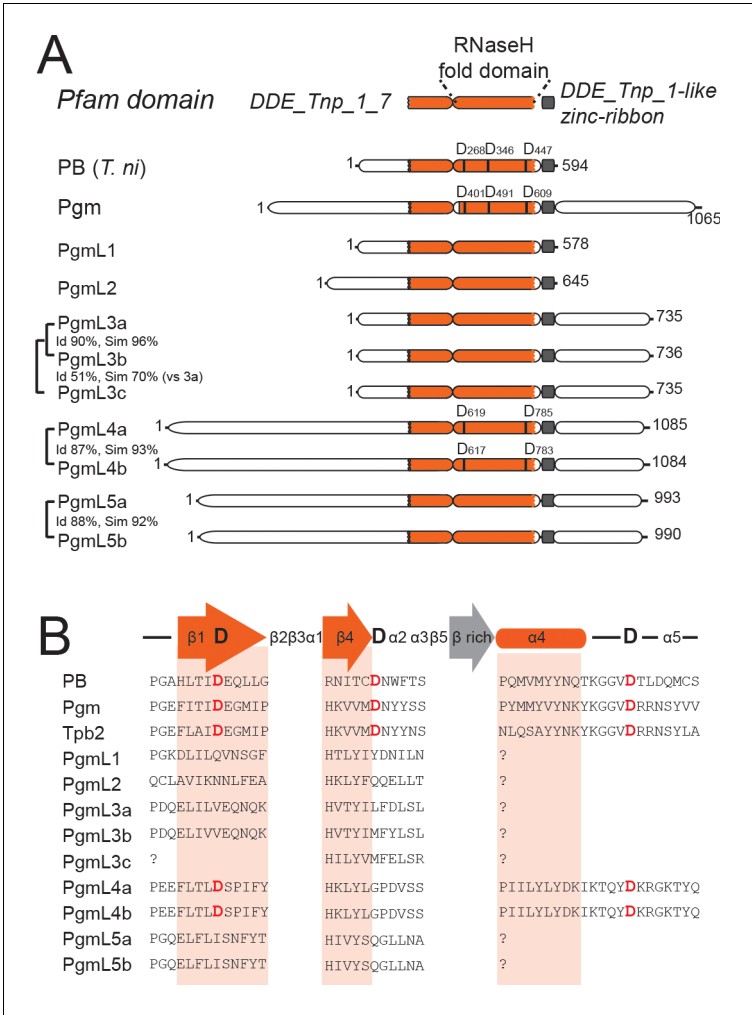

**Figure 1.** Novel domesticated PiggyBac transposases in *Paramecium*. (**A**) Domain organization of the PiggyBac transposase (PB) from *T. ni* and of *Paramecium* PiggyBac-related proteins (Pgm and PgmLs). The Pfam domain DDE_Tnp_1_7 is shown as a bipartite orange domain, with the RNase H fold corresponding to its right part (conserved catalytic D residues are indicated by vertical bars). The DDE_Tnp_1-like zinc ribbon is in grey. Id: % of amino acid identity; sim: % of similarity. (**B**) Protein sequence alignment of the residues surrounding the three catalytic aspartic acids (DDD). Following secondary structure prediction, sequence alignments were adjusted manually, using the expected position of the three catalytic D residues in the first and fourth β strands and immediately downstream of the fourth α helix of the RNase H fold domain, respectively (*Hickman et al., 2010*). '?' indicates that the expected α4 helix could not be predicted using the PSIPRED secondary structure prediction software.

DOI: https://doi.org/10.7554/eLife.37927.002

The following figure supplements are available for figure 1:

**Figure supplement 1.** MUSCLE alignment of the cysteine-rich domains of ciliate domesticated PB transposases and other PB transposases.

DOI: https://doi.org/10.7554/eLife.37927.003

**Figure supplement 2.** Maximum Likelihood tree of ciliate domesticated PB transposases and other PB transposases.

DOI: https://doi.org/10.7554/eLife.37927.004

PgmL1 and PgmL5a carboxy-terminal peptides (*Figure 2—figure supplement 1*). For PgmL2, PgmL3a and PgmL4a, transgenes expressing carboxy-terminal 3X Flag-tagged fusions under the control of their respective endogenous transcription signals were microinjected into the MAC of vegetative cells. Non-injected cells and transformants were grown then starved to induce autogamy.

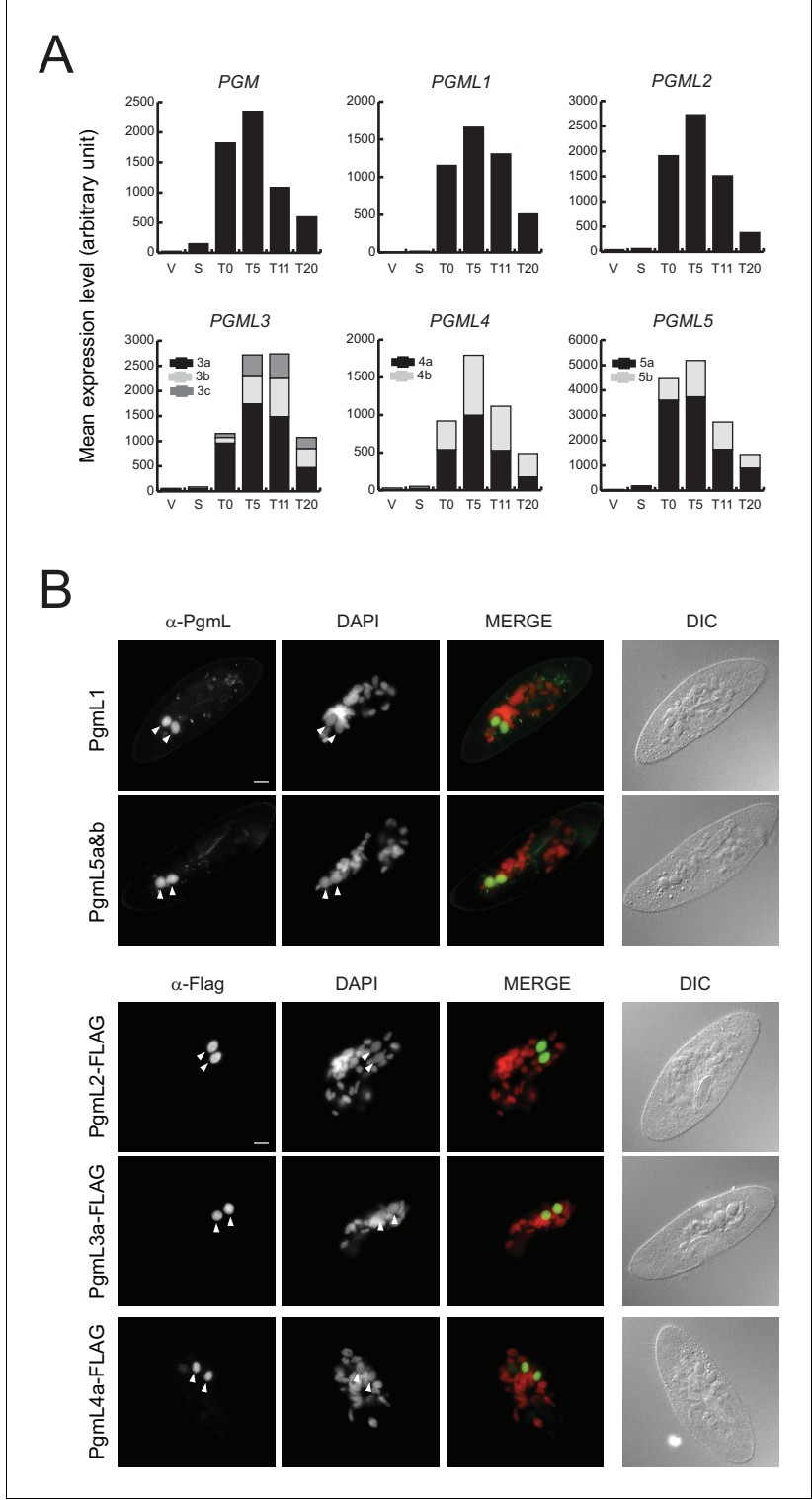

**Figure 2.** Expression and nuclear localization of PgmLs during autogamy. (**A**) Normalized RNA-seq data were extracted from (*Arnaiz et al., 2017*) and used to calculate mean expression levels for each time-point. V: vegetative cells (V1.2); S: starved cells with meiotic micronuclei (S1.1 and S1.2); T0: T0.1 and T0.2; T5: T5.1 and T5.2; T11: T11.1 and T11.2; T20: T20.1 and T20.2. All time-points are in hours. (**B**) Immunofluorescence staining of PgmL proteins in autogamous cells. White arrowheads point to developing new MACs. Scale bar: 10 μm.
DOI: https://doi.org/10.7554/eLife.37927.005

*Figure 2 continued on next page*

*Figure 2 continued*

The following figure supplements are available for figure 2:

**Figure supplement 1.** Validation of the specificity of antibodies directed against Pgm, PgmL1, PgmL5a, and the Flag peptide by immunofluorescence labelling of fixed cells.

DOI: https://doi.org/10.7554/eLife.37927.006

**Figure supplement 2.** Localization of GFP and RFP fusions in developing new MACs.

DOI: https://doi.org/10.7554/eLife.37927.007

Immunofluorescence microscopy allowed the detection of a specific signal in the developing MAC for all PgmLs 5 to 10 hr after the start of autogamy (*Figure 2B*). This stage corresponds to the time when Pgm appears in the new MACs (*Dubois et al., 2017*) and DNA cleavage takes place at IES ends (*Gratias and Bétermier, 2003*; *Gratias et al., 2008*; *Baudry et al., 2009*). Specific localization in the developing new MAC was confirmed using N-terminal GFP fusions for PgmL1, PgmL2 and PgmL5b, and a C-terminal RFP fusion for PgmL4a (*Figure 2—figure supplement 2*).

## Each *PGML* group is required for successful completion of autogamy

Functional analysis of *PGML* genes was performed by knocking down their expression using feeding-induced RNA interference (*Galvani and Sperling, 2002*). *PGML1*- or *PGML2*-knocked down cells were unable to produce viable post-autogamous progeny with a functional new MAC (*Figure 3A*, *Supplementary file 6*). For *PGML3* genes, specific silencing of *PGML3a* yielded only 30% viable sexual progeny, whereas no significant phenotype was observed following individual *PGML3b* or *PGML3c* silencing. In contrast, no sexual progeny were recovered in a double *PGML3a and b* KD, suggesting that the two paralogs have a redundant function. The contribution of *PGML3c* - the least expressed gene in the group - is unclear since knocking down this gene alone or together with *PGML3a* or *PGML3b* did not give a post-autogamous phenotype. Thus, even though *PGML3c* has been conserved in all *P. aurelia* species (*Supplementary file 2*), we cannot confirm that it carries out any important function. Similar results were obtained for *PGML4* and *PGML5* groups: knocking down both paralogs gave a stronger phenotype than individual silencing of each gene. We conclude that each *PGML* group as a whole is essential for the completion of autogamy and paralogs from the most recent WGD play redundant roles in the process.

## PgmLs can form complexes with Pgm

The *T. ni* PB transposase forms a dimer in solution and probably works as a higher-order oligomer during assembly of the transposition complex (*Jin et al., 2017*). Previous work in *Paramecium* established that Pgm multimerizes in cell extracts and several Pgm subunits are required to complete IES excision in vivo (*Dubois et al., 2017*). Like Pgm, PgmLs are essential for MAC development, even though they lack a complete DDD catalytic triad. We therefore considered the possibility that PgmLs interact with Pgm.

N-terminal HA-tagged versions of PgmL1, PgmL2, PgmL3a, PgmL4a and PgmL5a were expressed in insect cells using synthetic genes cloned into baculovirus vectors (*Supplementary file 5*). Soluble protein extracts were prepared from cells co-expressing each individual HA-fused PgmL with MBP-Pgm or MBP alone, and the ability of each PgmL to interact with Pgm was tested in MBP pull-down assays. Because recombinant MBP-Pgm binds DNA at low salt concentration (*Figure 3—figure supplement 2*), all assays were performed under high-salt conditions (500 mM NaCl) to avoid potential DNA-mediated interactions between proteins. We found that each HA-tagged PgmL co-precipitates with MPB-Pgm, whereas little or no co-precipitation is observed with MBP alone (*Figure 3B*). We confirmed the interaction between Pgm and each PgmL by showing that Pgm co-immunoprecipitates with HA-PgmLs using α-HA antibodies (*Figure 3—figure supplement 2*). These experiments demonstrate that PgmLs can form complexes with Pgm.

## *PGML* KDs compromise the correct nuclear localization of Pgm

The ability of each PgmL to interact with Pgm prompted us to check the fate of Pgm in PgmL-depleted cells. We knocked down each *PGML* group as a whole and the efficiency of each RNAi was attested by the absence of progeny with a functional new MAC (*Supplementary file 7*). For each

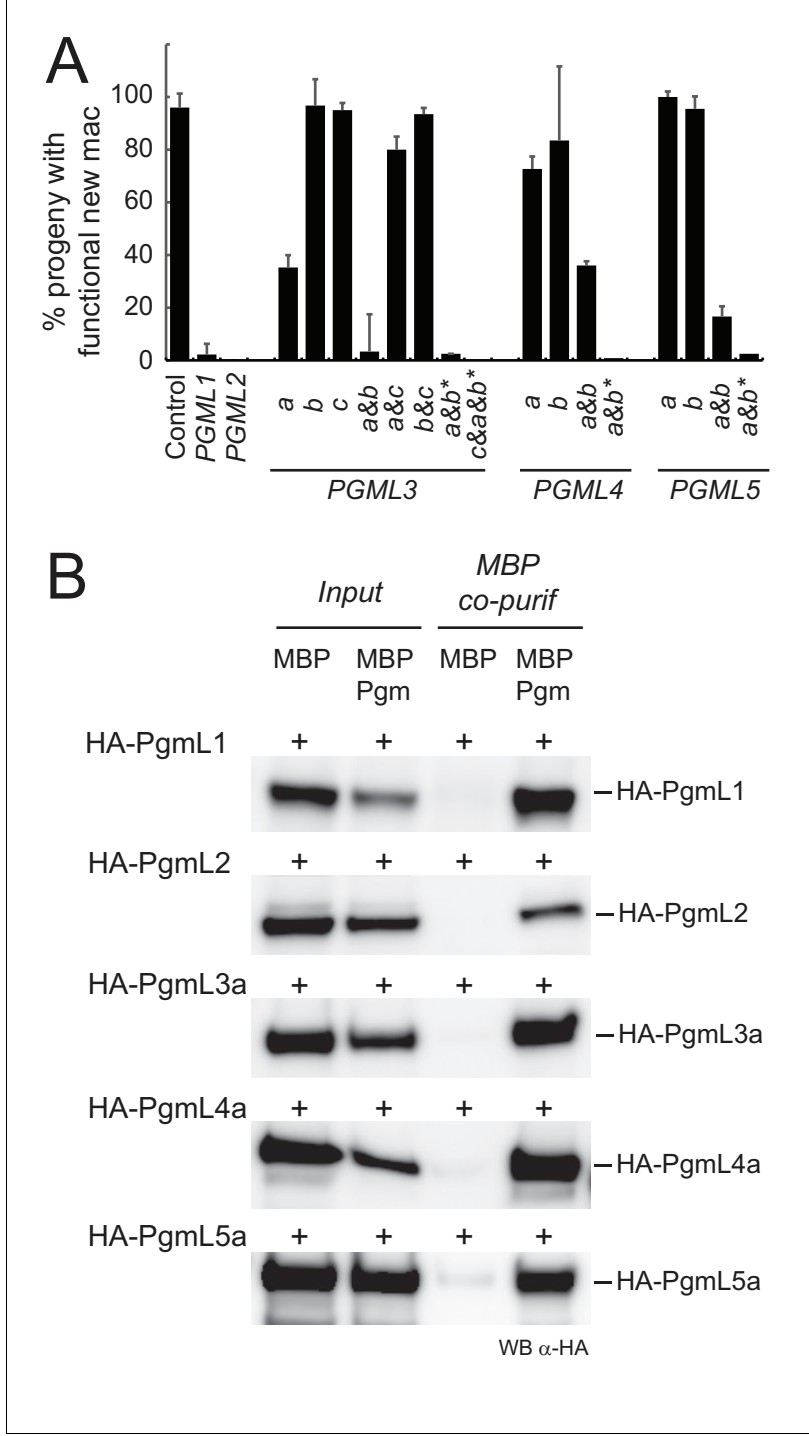

**Figure 3.** PgmLs are essential during autogamy and interact with Pgm in cell extracts. (**A**) Effect of *PGML* KDs on the recovery of post-autogamous progeny with functional new MACs. For *PGML1*, *PGML2* and *PGML3c,* only the results obtained using IF1 RNAi constructs (***Figure 3—figure supplement 1***) are shown. For groups of duplicated paralogs, individual gene KDs were performed using gene-specific IF2 constructs (***Figure 3—figure supplement 1***), while double KDs were performed using either IF2 or cross-hybridizing IF1 (*) constructs. Error bars represent standard deviations (n = 2 to 14, see ***Supplementary file 6***) (**B**) Pull down of HA-PgmL fusions with MBP-Pgm using recombinant proteins expressed in insect cells. In each panel, the HA-tagged protein that was co-expressed with MBP or MBP-Pgm is indicated on the left and the band revealed on western blots (WB) using anti-HA antibodies is indicated on the right. The full-size blot with molecular weight marker is shown in ***Figure 3—figure supplement 2***.

*Figure 3 continued on next page*

*Figure 3 continued*

DOI: https://doi.org/10.7554/eLife.37927.008

The following figure supplements are available for figure 3:

**Figure supplement 1.** Map and coordinates of *PGML* feeding inserts.

DOI: https://doi.org/10.7554/eLife.37927.009

**Figure supplement 2.** Co-precipitation of MBP-Pgm with HA-PgmL fusions.

DOI: https://doi.org/10.7554/eLife.37927.010

---

KD, autogamous cells were collected and fixed between T5 and T10, which corresponds to the time-window when the total cellular amount of Pgm is maximal in control cells (*Dubois et al., 2017*). Endogenous Pgm was monitored using immunofluorescence (*Figure 4A*) and immunoblotting (*Figure 4B*). We confirmed on western blots that Pgm is undetectable in Pgm-depleted cells (*Figure 4B*). No change in total cellular Pgm amounts was observed in PgmL-depleted cells relative to controls, indicating that neither Pgm expression nor stability are affected in *PGML* KDs.

Immunofluorescence staining, however, revealed that the endogenous nuclear Pgm signal is systematically lower in *PGML* KDs relative to control (*Figure 4A*). Quantification of the Pgm signal in new MACs revealed a 35% decrease in a *PGML1* KD and a ~ 75% decrease in every other *PGML* KD (*Figure 4C*), almost reaching the 85% decrease observed in a *PGM* KD (*Figure 4D*). Of note, the immunofluorescence protocol used here was set up for optimal detection of nuclear Pgm (*Dubois et al., 2017*) and includes a Triton-mediated permeabilization step prior to cell fixation (see Materials and methods). Because this pre-extraction procedure may affect the apparent localization of proteins that are not tightly held in the nucleus (*Martini et al., 1998*), we also performed Pgm immunostaining in *PGML* KDs omitting this step. Under these conditions, the quality of the control Pgm immunostaining was reduced and a higher background was observed, but we could still quantify the Pgm nuclear signal in the different KDs (*Figure 4—figure supplement 2*). We still observed a ≈35% decrease in *PGML1* KD relative to the control, and a 50% to 60% decrease in every other *PGML* KD. These data therefore indicate that *bona fide* Pgm nuclear localization is significantly affected by the depletion of PgmL2, PgmL3, PgmL4 or PgmL5 and, to a lesser extent, by the depletion of PgmL1. The significant exacerbation of the localization defect observed in PgmL2, PgmL3, PgmL4 or PgmL-5-depleted cells subjected to a pre-extraction procedure further indicates that depletion of these particular PgmL reduces the strength of Pgm association with the nucleus (*Figure 4—figure supplement 2*).

## *PGML* KDs have a genome-wide impact on IES elimination

To gain genome-wide insight into the effect of *PGML* KDs on IES excision, large-scale cultures were subjected to RNAi against *PGML1*, *PGML2*, *PGML3a and b*, *PGML4a and b* or *PGML5a and b*, and genomic DNA was extracted from isolated nuclei at late autogamy stages for high-throughput sequencing (*Supplementary file 8*). IES retention scores (IRS) were computed for each sample, using IES$^+$ sequencing reads matching IES boundaries and IES$^-$ reads matching precise IES excision junctions (*Denby Wilkes et al., 2016*, see Materials and methods). The efficiency of *PGML* KDs was checked using northern blot hybridization of total RNA from autogamous cells (*Figure 5—figure supplement 1*) and confirmed by the absence of viable post-autogamous progeny (*Supplementary file 7*).

The distributions of IRS show that every *PGML* KD strongly inhibits IES excision genome-wide (*Figure 5A*). Differences, however, are observed between the five *PGML* groups. *PGML2*, *PGML4a and b* or *PGML5a and b* KDs result in significant retention of all IESs in the new MAC. *PGML1* and *PGML3a and b* KDs still allow efficient excision of a fraction of IESs, referred to as non-significantly retained (*i.e.* excised) following statistical analysis of IRS (*Denby Wilkes et al., 2016*): this represents 7479 and 3511 IESs, respectively. Strikingly, 89% of excised IESs in a *PGML3* KD are also excised in a *PGML1* KD (*Figure 5B*). Of note, excised IESs in *PGML1* or *PGML3a and b* KDs tend to be significantly longer (median size = 62 or 67 bp, respectively) than an equivalent number of strongly retained IESs in the same KDs (median size = 48 or 55 bp, respectively) (*Figure 5C*). Analysis of the size distributions reveals that the size bias can mainly be attributed to over-representation of IESs

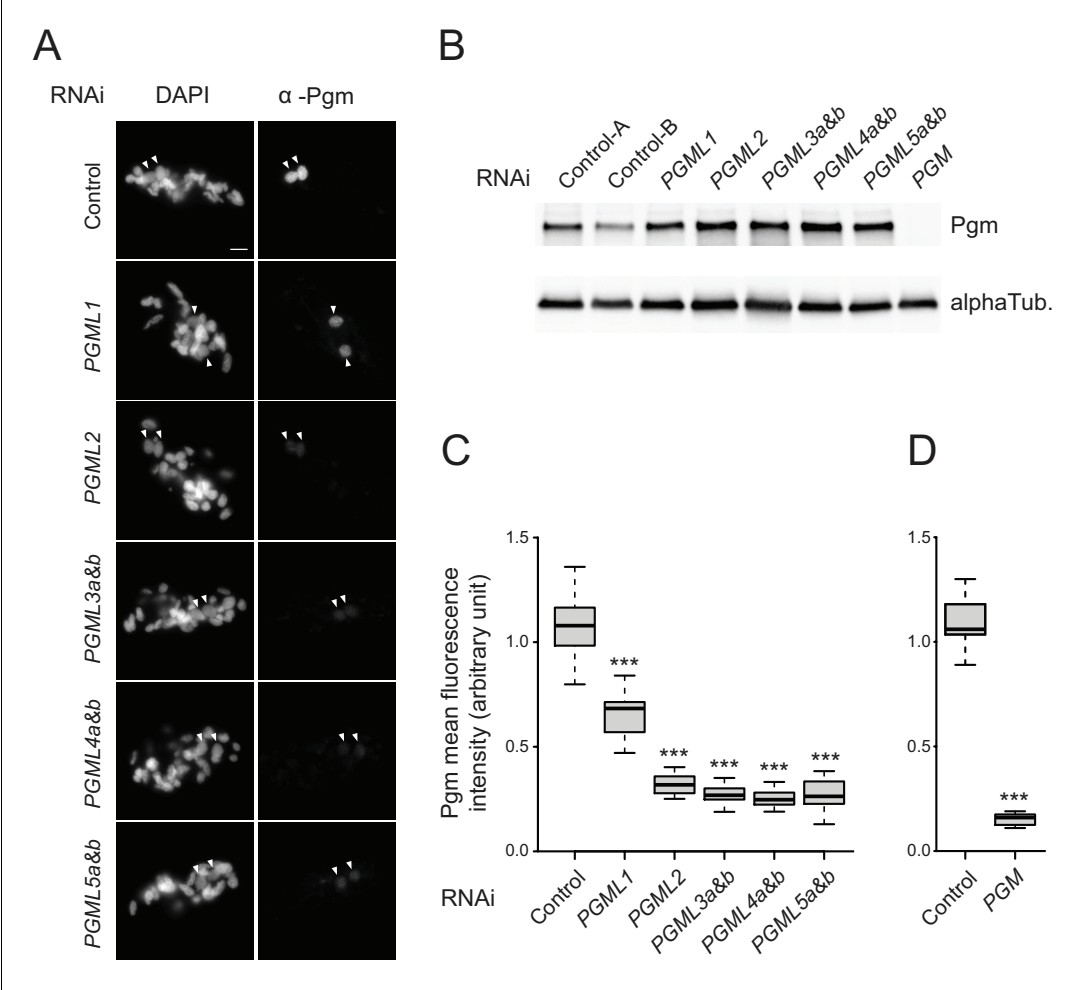

**Figure 4.** Expression and localization of Pgm in *PGML* KDs. (**A**) Immunostaining of Pgm in early autogamous cells subjected to control (L4440) or *PGML* RNAi. Developing MACs are indicated by white arrowheads. Scale bar is 10 μm. (**B**) Western blot analysis of Pgm expression in early autogamous cells subjected to control (L4440: two independent controls A and B are shown), *PGML* or *PGM* RNAi. (**C**) Boxplot representation of the distribution of Pgm fluorescence intensities quantified in 30–55 μm² developing MACs subjected to the different RNAi shown in (**A**). This size window corresponds to the maximal Pgm signal in the control (*Figure 4—figure supplement 1*) and was chosen to quantify nuclear Pgm immunofluorescence for all KDs, since no significant size difference was noticed for developing MACs relative to the control. For each condition, 19 to 35 developing MACs were analyzed. (**D**) Independent set of experiments showing the quantification of Pgm fluorescence intensity in 30–55 μm² developing MACs following control (*ND7*) or *PGM* RNAi. 11 and 12 MACs were analyzed, respectively. In (**C**) and (**D**): *** for p<0.001 in a Mann-Whitney-Wilcoxon statistical test (see Materials and methods for details).

DOI: https://doi.org/10.7554/eLife.37927.011

The following figure supplements are available for figure 4:

**Figure supplement 1.** Plot of Pgm mean immunofluorescence intensity *vs* developing MAC size in cells subjected to control or *PGML* RNAi.

DOI: https://doi.org/10.7554/eLife.37927.012

**Figure supplement 2.** Immunolocalization of Pgm without Triton extraction in *PGML* knockdowns.

DOI: https://doi.org/10.7554/eLife.37927.013

from the 75 to 77 bp peak (and larger sizes) (*Figure 5D*). Under-representation of 55 to 57 bp IESs can also be noted.

To investigate whether the size biases described above are a specificity of *PGML1* and *PGML3a and b* KDs or simply attributable to incomplete RNAi, we partially released *PGML2* KD by diluting *PGML2* RNAi-inducing medium in RNAi medium targeting a non-essential gene (see *Dubois et al., 2017*). Consistent with higher survival in the progeny (*Supplementary file 7*), partial *PGML2* KD (1:4

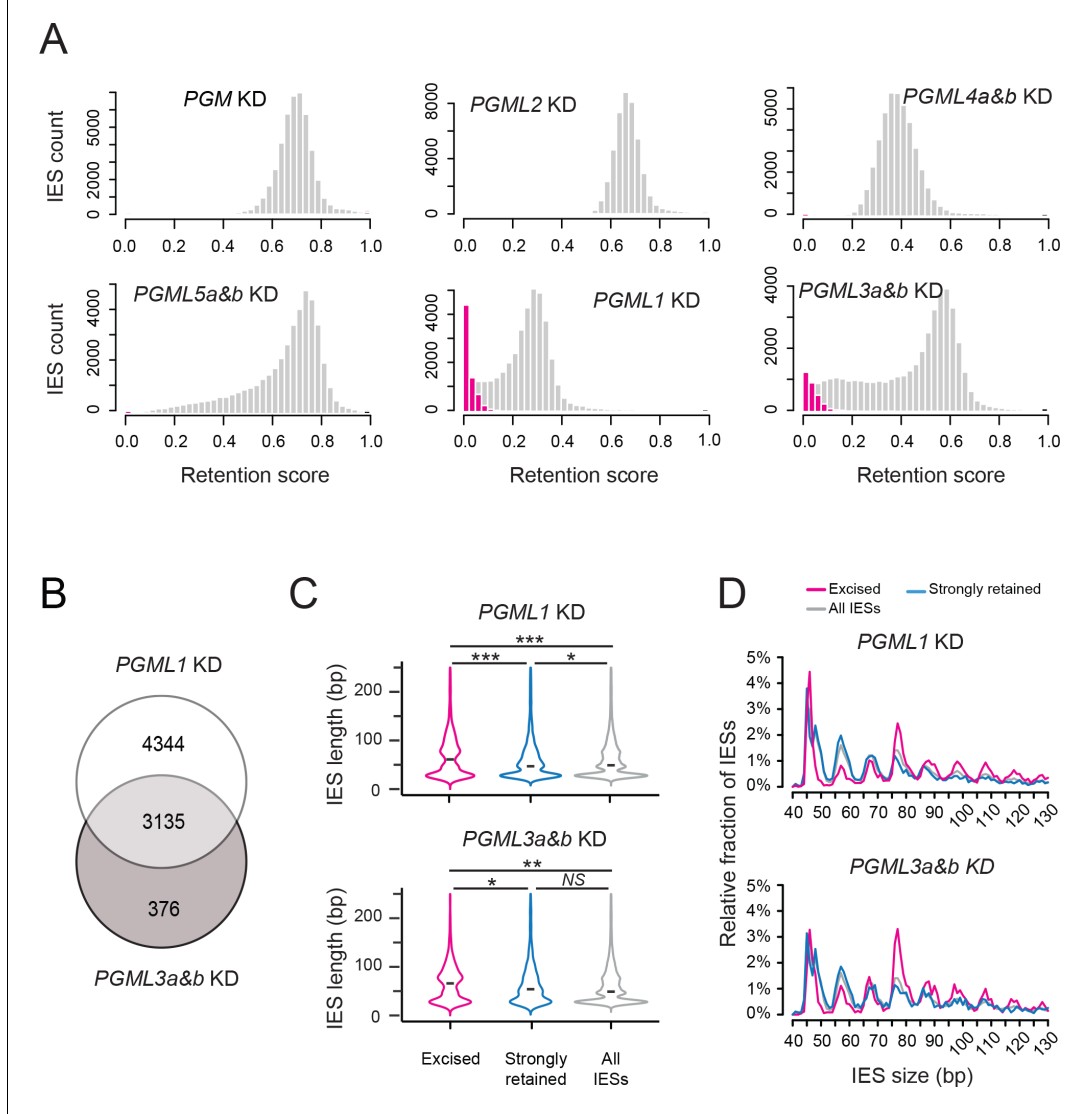

**Figure 5.** Analysis of IES retention in *PGML* KDs. (**A**) Distribution of IES retention scores (IRS) in *PGML* KDs. Grey bars represent the distribution of all IESs over IRS ranging from 0 to 1 (by bins of 0.025). The distribution obtained in a previously published *PGM* KD (***Arnaiz et al., 2012***) is shown as a control. Absolute values of IRS should not be compared from one KD to the other, due to variable contamination by old MAC fragments. For *PGML1* and *PGML3a and b* KDs, the distribution of statistically non-significantly retained IESs (*i.e.* excised IESs) is superimposed in magenta. (**B**) Venn diagram representing the overlap between the sets of excised IESs in *PGML1* or *PGML3a* and *b* KDs (**C**) Violin plots of IES length distributions for the population of non-significantly retained IESs (magenta; n = 7479 in *PGML1* KD and n = 3511 in *PGML3a and b* KD), the same number of IESs with the highest retention scores (blue) and all IESs (grey). The black dash shows the median of each distribution. Plots were drawn using the ggplot2 R-package (***Wickham, 2009***). Size distributions were compared using a Mann-Whitney-Wilcoxon statistical test and p values are indicated for each comparison (***: $p<2.2 \ 10^{-16}$; **: $2.2 \ 10^{-16}<p<10^{-10}$; *: $10^{-10}<p<5 \ 10^{-2}$; NS: $p>5.5 \ 10^{-2}$) (**D**) Comparative analysis of the relative fraction of IESs in each size peak among the populations of excised (magenta) and strongly retained (blue) IESs in *PGML1* and *PGML3a and b* KDs, and the whole IES population (grey). Only IESs between 40 and 130 bp are represented.

DOI: https://doi.org/10.7554/eLife.37927.014

The following figure supplements are available for figure 5:

**Figure supplement 1.** Northern blot analysis of *PGML* mRNA during autogamy in *PGML* knockdowns.
DOI: https://doi.org/10.7554/eLife.37927.015

**Figure supplement 2.** Analysis of IES retention scores in partial *PGML2* KDs.
DOI: https://doi.org/10.7554/eLife.37927.016

**Figure supplement 3.** Correlation between IES retention scores in *PGM*, *PGML* and partial *PGML2* knockdowns.
DOI: https://doi.org/10.7554/eLife.37927.017

dilution) shifts the distribution of IRS toward zero and allows efficient excision of a fraction of 10,297 IESs (*Figure 5—figure supplement 2*). The distributions of IRS in partial *PGML2* KD correlate with those obtained in *PGML1* or *PGML3* KDs (Spearman's rank correlation coefficients = 0.7 and 0.77, respectively; *Figure 5—figure supplements 2* and *3*), indicating that a similar gradient is overall established from excised to strongly retained IESs in these KDs. However, comparison of the size distributions of excised versus strongly retained IESs in the three conditions confirms that over-representation of large IESs (75 to 77 bp in length and larger) is specific to excised IESs in *PGML1* and *PGML3* KDs (*Figure 5D*, *Figure 5—figure supplement 2*).

## PgmL1- or PgmL3a and b-depleted cells are prone to IES excision errors

Previous whole-genome DNA sequencing of wild-type reference strains revealed that IES excision generates sequence heterogeneity in the somatic genome (*Duret et al., 2008*; *Swart et al., 2014*), most likely due to erroneous excision events taking place between two TA dinucleotides, one of which at least is localized at an alternative position relative to reference IES boundaries. To evaluate the background of IES excision errors in the absence of any KD, we first sequenced total DNA extracted from parental vegetative cells before they were subjected to *PGML* RNAi. We analyzed sequencing reads mapping to the MAC + IES reference and counted the number of erroneous excision reads present in each sample. Consistent with published data, we found a low number of erroneous junctions (2.4% to 3% of all excision reads) in vegetative MACs (V) formed under no-KD conditions (*Supplementary file 9*).

We then sequenced genomic DNA of nuclei-enriched preparations from autogamous cells originating from the parental cultures described above and subjected to *PGML* RNAi. As in wildtype vegetative MACs, a fraction of excision reads (2.9% to 3.7%), representing the contribution of both the new MAC and old MAC fragments, correspond to erroneous junctions. For each *PGML* KD, we calculated a normalized number of de novo excision errors that are specific to the new MAC (Materials and methods). Fewer de novo errors (26 to 38 per million mapped reads) are observed in *PGML2*, *PGML4* or *PGML5* KDs than in no-KD controls (*Figure 6A*), consistent with our observation that no significant excision activity is detected in these *PGML* KDs. In contrast, *PGML1* and *PGML3* KDs yield similar de novo excision counts (122 and 98, respectively) relative to control (*Figure 6A*), in spite of strongly reduced IES excision activity (*Figure 5A*), suggesting that residual excision in these KDs tends to be error-prone. The observation that the fraction of IESs, for which at least one error is detected in *PGML1* or *PGML3a and b* KDs, is higher among excised IESs provides support to this idea: specifically for these two KDs, the fraction of fully excised IES (IRS < 0.025) with errors reaches ~45%, *versus* ~10% under other conditions (*Figure 6—figure supplement 1*).

Erroneous junctions can be grouped in different classes according to the location of the TAs used as alternative excision boundaries (*Figure 6B*). We found that *PGML1* or *PGML3* KDs modify the relative proportions of de novo error classes compared to a no-KD control or to *PGML2*, *PGML4* or *PGML5* KDs (*Figure 6C* and *Supplementary file 9*). The most conspicuous change is a specific increase in the proportion of partial internal excision errors, reaching more than 50% of all de novo errors in PgmL1- or PgmL3-depleted cells. Over-representation of this particular class of errors is not observed in partial *PGML2* KDs (*Figure 6—figure supplement 2*), indicating that erroneous targeting of alternative internal boundaries is not a general consequence of limiting availability of the excision machinery, but is a specificity of *PGML1* and *PGML3* KDs. In the latter two KDs, most erroneous internal boundaries are shifted by 10 to 11 bp from the canonical TA, resulting in excision of DNA fragments shortened by one helical turn (*Figure 6D*). The position of alternative boundaries cannot be explained simply by a biased distribution of TA dinucleotides in IESs, since internal TAs can be found 5 to 6 bp away from reference IES boundaries (*Figure 6—figure supplement 3*), but they are not used in partial internal excision errors. Except for the TA, erroneous alternative boundaries do not match the general consensus sequence for IES ends (5'-<u>TA</u>YAGYNR-3'), nor do they define a novel sequence motif (*Figure 6—figure supplement 3*). Moreover, 'unused' canonical ends exhibit no particular sequence motif - different from the general consensus - that would suggest a preference of PgmL1 or PgmL3 for a specific sequence. Finally, we noticed that error-prone IESs, for which an internal TA (mostly shifted by 10 to 11 bp) is used in erroneous excision events, follow a different size distribution from the global IES population (*Figure 6E* and *Figure 6—figure supplement 4*): IESs of 66 bp and above are over-represented, whereas 47 bp and shorter IESs are largely under-

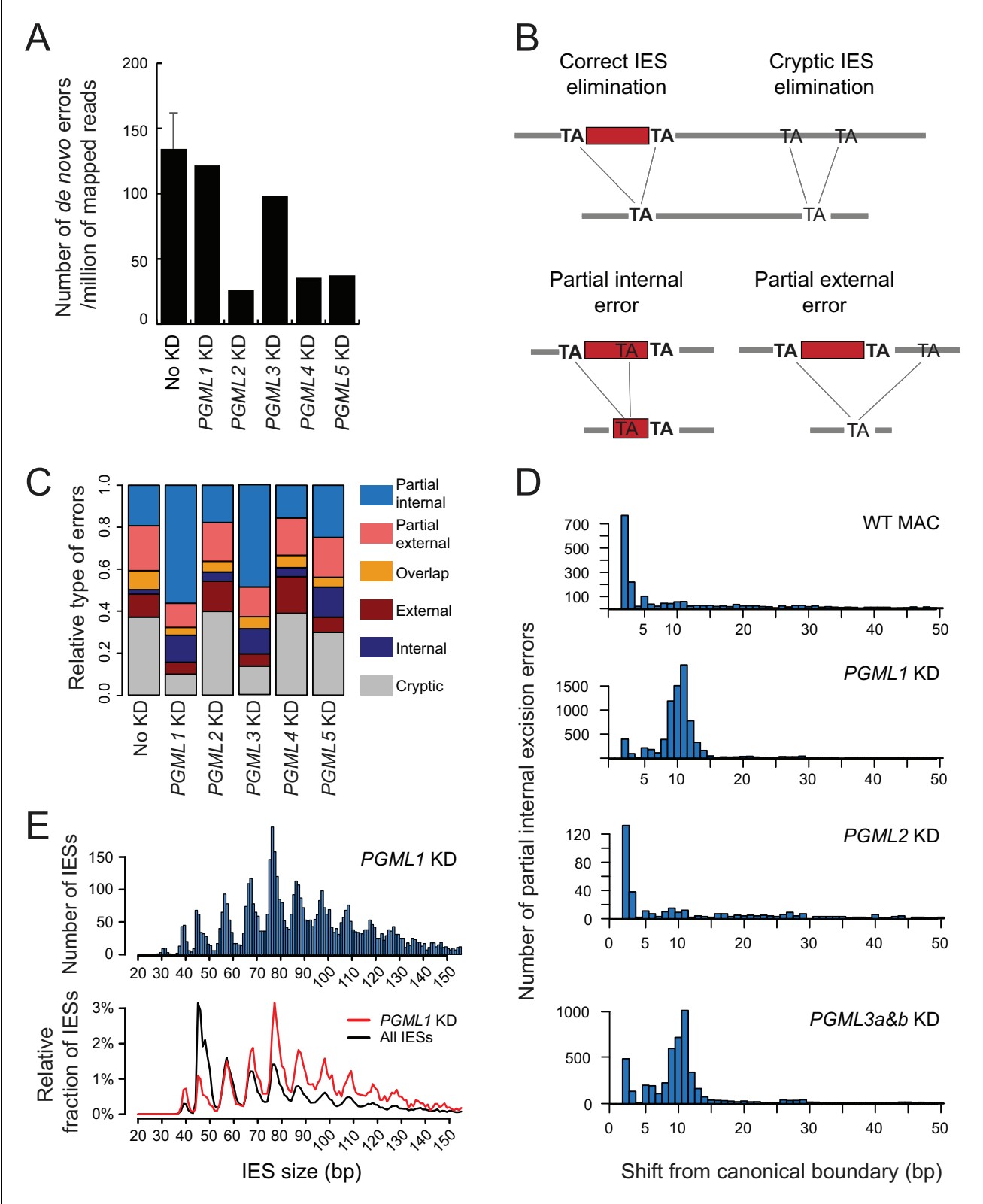

**Figure 6.** IES excision errors in *PGML* KDs. (**A**) Number of IES excision errors in the MAC of vegetative cells before autogamy (No KD) and in the new MAC of autogamous cells upon each *PGML* KD (de novo errors). For the No KD sample, the error bar represents the standard deviation for five replicates (V samples in *Supplementary file 9*). (**B**) Major classes of IES excision errors found in the different samples. In external or internal errors, the two alternative TAs are misplaced (one on each flanking side or both inside of a reference IES, respectively). Overlapping errors use one TA inside and

*Figure 6 continued on next page*

*Figure 6 continued*

the other outside of reference IESs. (C) Distribution of the different classes of de novo excision errors in *PGML* KDs. As a control, the distribution of pre-existing errors found in the old MAC is shown for a vegetative culture (see **Supplementary file 9**). (D) Position of alternative excision boundaries used in partial internal excision errors, relative to the canonical boundary of the reference IES. WT: vegetative MAC; for *PGML* KDs, only de novo errors were considered. (E) Size distribution of IESs exhibiting partial internal errors in a *PGML1* KD. Upper panel: size distribution of all IESs with partial internal errors. Lower panel: the black curve shows the fraction of IESs of each size relative to the total number of IESs in the genome; the red curve shows the fraction of IESs of each size among the population of IESs showing at least one partial internal error in a *PGML1* KD. In both panels, only IESs with an alternative boundary at >2 bp from the canonical one were counted. In the bottom panel, IESs shorter than 35 bp were not considered (see **Figure 6—figure supplement 4**).

DOI: https://doi.org/10.7554/eLife.37927.018

The following figure supplements are available for figure 6:

**Figure supplement 1.** The number of excision errors increases for IESs with the lowest retention scores in *PGML1* or *PGML3a and b* knockdowns.

DOI: https://doi.org/10.7554/eLife.37927.019

**Figure supplement 2.** Raw counts of IES excision errors in *PGML1*, *PGML3a and b* and partial *PGML2* knockdowns.

DOI: https://doi.org/10.7554/eLife.37927.020

**Figure supplement 3.** Alternative excision boundaries used in partial internal IES excision errors.

DOI: https://doi.org/10.7554/eLife.37927.021

**Figure supplement 4.** Size distribution of IESs with partial internal excision errors in *PGML1* or *PGML3a* and *b* KDs.

DOI: https://doi.org/10.7554/eLife.37927.022

represented. This size bias is similar to the distribution of excised IESs in *PGML1* or *PGML3* KDs, indicating again that errors are linked to excision activity.

## Discussion

### IES excision in *Paramecium* is carried out by multiple partners including Pgm and five novel domesticated PB transposases

The present discovery of novel essential Pgm partners, encoded by five groups of paralogous genes and collectively referred to as PgmL1 to PgmL5, brings new insight into the mechanism of IES excision and deeper understanding of the molecular machinery involved. PgmLs are domesticated PB transposases distantly related to Pgm. Consistent with their essential function, Pgm and PgmLs are conserved within the *P. aurelia* group as well as in the more distant *P. caudatum*, indicative of an ancient origin of the IES excision machinery in *Paramecium*. More work is clearly needed to unravel the nature and organization of the IES excision machinery, but our data are in favor of a unique protein complex embedding at least two Pgm subunits and one PgmL subunit from each group (**Figure 7**). Indeed, depletion of each single PgmL group is sufficient to strongly compromise the nuclear localization of Pgm. As a consequence, each *PGML* KD inhibits excision of a large majority of IESs (83% to 100%) genome-wide. The remaining ~17% IESs that are still excised in *PGML1* or *PGML3* KDs are prone to excision errors, suggesting that PgmLs are stricty required for efficient and accurate excision.

We show here that PgmLs can each interact directly with Pgm and we previously reported that Pgm also has homo-oligomerization properties (**Dubois et al., 2017**). Future studies will address whether these interactions are mutually exclusive, whether PgmLs interact with each other and, more importantly, which among these interactions participate in the assembly of the full excision complex. The *Paramecium* system shares interesting features with the higher-order complexes (transpososomes) that interact with transposon ends during *bona fide* transposition. Biochemical and structural studies of DD(D/E) transposases and retroviral integrases bound to their cognate DNA substrates have revealed that assembly of a productive transpososome often involves more transposase subunits than those actually engaged in catalysis, the supernumerary subunits playing an architectural role within the transposition complex (**Montaño et al., 2012**; **Hickman et al., 2014**); reviewed in **Hickman and Dyda, 2015**). In particular, recent studies have shown that the *T. ni* PB transposase dimerizes in solution (**Jin et al., 2017**) and higher-order complexes were proposed to assemble during *piggyBac* transposition (**Morellet et al., 2018**). Within a transpososome, all transposase subunits are generally identical because they are encoded by a single gene carried by the mobile element itself. In *Paramecium*, multiple domesticated transposase genes encode the

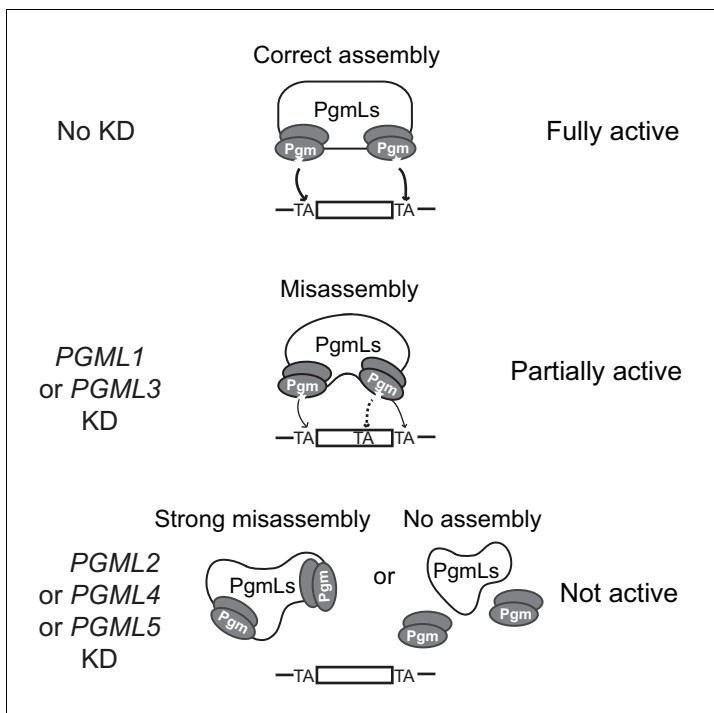

**Figure 7.** Model for IES excision mediated by a multicomponent Pgm/PgmLs complex. This figure summarizes the observed effects of *PGML* KDs on Pgm-mediated IES excision. In line with previously published data (*Dubois et al., 2017*) and known properties of the *T. ni* PiggyBac transposase (*Jin et al., 2017*), the catalytically active form of Pgm is assumed to be a dimer. In the absence of information about the stoichiometry of the complex, one Pgm homodimer (active catalytic site drawn as a star) is represented at each IES boundary, with a large bridging structure formed by all PgmLs. In a fully assembled complex, PgmL subunits are proposed to drive the correct positioning of the Pgm catalytic site onto both TA cleavage sites (indicated by arrows). Following *PGML* KDs, we propose two distinct situations. In *PGML1* or *PGML3* KDs, the depleted complexes can exist but are misassembled. As a consequence, Pgm nuclear stability is reduced (the phenotype is more pronounced in a *PGML3* KD than in a *PGML1* KD) and Pgm activity is altered, because incorrect positioning of catalytic subunits generates specific partial internal excision errors at low frequency (a dotted arrow points to the erroneously targeted alternative TA). In the other three KDs, IES excision complexes depleted for PgmL2, PgmL4 or PgmL5 are totally inactive. This might result either from strong misassembly of the complex or its dissociation (or non-assembly).

DOI: https://doi.org/10.7554/eLife.37927.023

different subunits that together carry out IES elimination. Selection pressure to maintain a fully conserved catalytic triad has been exerted on only one of these genes (*PGM*) and previous work established that at least two Pgm subunits are present in the complex (*Dubois et al., 2017*), consistent with a model in which their catalytic sites are positioned on each TA boundary (*Figure 7*). The catalytic domain of PgmL proteins has evolved more rapidly, suggesting that PgmLs rather have an architectural function within the complex. Moreover, PgmL proteins differ in their domain organization and PgmL depletions have different effects on IES excision efficiency and accuracy, which suggests that all PgmLs may not play exactly the same role. Of note, the conservation of two Ds in PgmL4, which is the most closely related to Pgm, indicates that its RNase H domain, although probably inactive, has evolved under selection pressure, suggesting that it may play a particular role in the catalysis of DNA cleavage. RNAi-mediated depletion of PgmL2, PgmL4 or PgmL5 completely abolishes Pgm cleavage activity and its stable nuclear localization, perhaps by impairing correct assembly of the excision complex (*Figure 7*) and/or loosening its interaction with chromatin, preventing us at this stage from proposing a specific function for these three proteins. In contrast, PgmL1- or PgmL3-depleted complexes exhibit residual activity associated with an increased bias for partial internal excision errors involving the use of alternative 10 bp shifted TA boundaries. Because such a bias is not observed in partial *PGML2* KDs, our data cannot simply be attributed to partial depletion and

rather point to a specific function of PgmL1 and PgmL3. We propose that incomplete machineries devoid of PgmL1 or PgmL3 can still interact with IESs (*Figure 7*), even though much less efficiently and with lower accuracy than fully assembled complexes. The size biases observed in the population of excised IESs in *PGML1* or *PGML3* KDs might reveal that one function of these two PgmL subunits is to provide the architectural versatility required to adjust to the variable features of eliminated sequences.

## Size biases of partial internal excision errors recapitulate evolution-driven IES size distribution

*Paramecium* IESs exhibit a characteristic size distribution, with a minimum size of 26 bp and a ~ 10 bp periodicity proposed to result from mechanistic constraints on IES excision (*Arnaiz et al., 2012*). The present study of partially active PgmL1- or PgmL3-depleted complexes is consistent with this hypothesis.

The overlapping subsets of IESs that still excise efficiently in *PGML1* and *PGML3a and b* KDs tend to be larger in size than an equivalent pool of strongly retained IESs, with an under-representation of 46 to 47 bp IESs and over-representation of 75 to 77 bp (and larger) IESs. No specific sequence other than the usual consensus was found at the ends of excised or strongly retained IESs in these KDs, indicating that no particular motif defines the subsets of PgmL1- (or PgmL3)-dependent or independent IESs. We propose, instead, that PgmL1 and PgmL3 contribute to the positioning of Pgm-dependent DNA cleavages at correct TA boundaries, thus fine-tuning the size of the excised sequences (*Figure 7*). Indeed, residual activity of PgmL1- or PgmL3-depleted machineries reveals an over-representation of partial internal excision errors, leading to excision of IES-derived fragments one turn of a DNA helix shorter than reference IESs, a distance that corresponds to the periodicity of the IES size distribution. Remarkably, erroneous excision of 46 to 47 bp IESs that would have led to excision of 36 to 37 bp fragments is not observed (*Figure 6E*). Furthermore, 36 to 37 bp IESs are prone to partial internal errors resulting in excision of 26 to 28 bp fragments. These observations indicate that sequences of 36–37 bp may be mechanistically difficult to excise, as proposed previously based on the existence of a 'forbidden' peak corresponding to this size range in the distribution of genomic reference IESs (*Arnaiz et al., 2012*). Likewise, IESs from the 26 to 28 bp peak do not yield erroneously excised 10 bp shorter TA-indels, supporting the notion that 26 bp is the minimum size for excision by the Pgm-associated machinery. Taken together, our data provide strong experimental support to the hypothesis that mechanistic limitations have imposed strong constraints on *Paramecium* IES size during evolution.

## Domesticated PB transposases in ciliates and other species

The distantly related ciliate *Tetrahymena thermophila*, which separated from *Paramecium* at least ~500 M years ago (*Parfrey et al., 2011*), harbors a clear Pgm ortholog, Tpb2p (*Figure 1—figure supplement 2*) that carries out imprecise excision of intergenic IESs, which constitute the vast majority of IESs in this ciliate (*Cheng et al., 2010*; *Vogt and Mochizuki, 2013*; *Hamilton et al., 2016*). Additional *Tetrahymena* domesticated PB transposases (Tpb7p and Lia5p), also lacking a conserved DDD triad, may somehow be related to *Paramecium* PgmLs, although the evolutionary relationships between *Tetrahymena* and *Paramecium* proteins are difficult to assess (*Figure 1—figure supplement 2*). While the role of Tpb7p is unknown (*Cheng et al., 2016*), Lia5p is essential for Tpb2p-dependent DNA elimination, localizes on IESs before excision and is involved in the delimitation of their excision boundaries (*Shieh and Chalker, 2013*; *Suhren et al., 2017*). Lia5p and Tpb7p may represent functional homologs of PgmLs for IES excision, but whether they interact with Tpb2p and how they contribute to DNA elimination at the molecular level remain to be established. In addition to the Tpb2p/Lia5p system, *Tetrahymena* encodes two other domesticated PB transposases, Tpb1p and Tpb6p, that precisely excise a distinct subset of 12 intragenic *piggyBac*-related IESs, in a Tpb2p-independent manner (*Cheng et al., 2016*; *Feng et al., 2017*). Thus, in contrast to *Paramecium*, *Tetrahymena* possesses two distinct IES excision machineries, each responsible for elimination of a particular subset of IESs. In spite of their differences, the two ciliate species provide remarkable examples of the participation of multiple-component protein complexes composed of catalytic and non-catalytic subunits, all domesticated from the same transposase family, in a transposition-related reaction. In humans, the Pgbd1 to Pgbd5 PB domesticated transposases do not all carry a fully

conserved catalytic triad. Based on our *Paramecium* work, future investigations should take into consideration the possibility that Pgbd proteins may be involved together in the same cellular function(s).

## Materials and methods

### In silico protein sequence analysis

*Paramecium* genes and protein sequences were uploaded from the ParameciumDB database (*Arnaiz and Sperling, 2011*) (https://paramecium.i2bc.paris-saclay.fr/) and accession numbers are displayed in *Supplementary file 2*. HMMer version 3.1b (http://eddylab.org/software/hmmer3/3.1b2/Userguide.pdf, default parameters) was used to search for Pgm-related proteins using the DDE_Tnp1_7 domain (PF13843) as the query and the predicted *Paramecium* proteins from ParameciumDB (v1.77) as the database. Multiple protein sequence alignments were performed using MUSCLE (http://www.ebi.ac.uk/Tools/msa/muscle/) (*Edgar, 2004*). Secondary structures were predicted using PSIPRED (V3.3) at the UCL website (http://bioinf.cs.ucl.ac.uk/psipred/) (*Jones, 1999*).

### *Paramecium* strains and standard culture conditions

*P. tetraurelia* wild-type 51 new (*Gratias and Bétermier, 2003*) or its mutant derivative 51 *nd7-1* (*Dubois et al., 2017*) were grown in a standard medium made of a wheat grass infusion inoculated with *Klebsiella pneumoniae* and supplemented with β-sitosterol (0.8 µg/mL) (*Beisson et al., 2010*). Autogamy was carried out as described (*Dubois et al., 2017*).

### Gene knockdown experiments

RNAi during autogamy was achieved using the feeding procedure, as described (*Dubois et al., 2017*). Briefly, *Paramecium* cells grown for 10 to 15 vegetative fissions in plasmid-free *Escherichia coli* HT115 bacteria (*Timmons et al., 2001*) were transferred to medium containing non-induced HT115 harboring each RNAi plasmid and grown for ~4 divisions. Cells were then diluted into plasmid-containing HT115 induced for dsRNA production and allowed to grow for ~8 additional vegetative divisions before the start of autogamy. Final volumes were 3 to 4 mL for small-scale experiments, 50 to 75 mL for middle-scale experiments, 4 L for large-scale experiments. The presence of a functional new MAC in the progeny was tested after four days of starvation, as described (*Dubois et al., 2017*).

Control experiments were performed using the L4440 vector (*Kamath et al., 2001*) or plasmid p0ND7c, which targets RNAi against the non-essential *ND7* gene (*Garnier et al., 2004*). For *PGML* KDs, PCR fragments from each *PGML* gene (*Figure 3—figure supplement 1*) were inserted into the multiple cloning site of L4440. Two different RNAi-inducing constructs targeting distinct non-overlapping regions (IF1 and IF2) were used for each gene: within multi-gene *PGML* families, IF1 shared strong nucleotide sequence homology between paralogs, while IF2 regions were gene-specific.

### RNA and DNA extraction

During autogamy, total RNA was Trizol-extracted from ~2 to $5 \times 10^5$ cells for each time-point and quantified using a NanoDrop spectrophotometer. Gel electrophoresis and northern blot hybridization with $^{32}$P-radiolabelled DNA probes were performed as described (*Baudry et al., 2009*). For PCR analysis, total genomic DNA was extracted from ~1 to $3 \times 10^3$ cells for each time-point using the NucleoSpin Tissue extraction kit (Macherey Nagel). For high throughput DNA sequencing, genomic DNA was extracted from whole cells (*Gratias and Bétermier, 2003*; *Arnaiz et al., 2012*) or isolated nuclei (*Gratias and Bétermier, 2003*; *Arnaiz et al., 2012*).

### Microinjection of transgenes expressing tagged PgmL proteins

Plasmids expressing PgmL2-3X Flag, PgmL3a-3X Flag and PgmL4a-3X Flag are pUC18 derivatives, in which the *P. tetraurelia PGML2, PGML3a* and *PGML4a* genes, respectively, were fused at their 3' end to a synthetic DNA sequence (Integrated DNA Technologies) encoding the 3X Flag tag (YKDHDGDYKDHDIDYKDDDDKT). All sequences are available upon request. Each transgene-bearing plasmid was linearized with *Bsa*I and microinjected with an *ND7*-complementing plasmid into the MAC of vegetative 51 *nd7-1* cells, as described (*Dubois et al., 2017*).

## Immunofluorescence and western blot analysis

Polyclonal α-Pgm 2659-GP guinea pig antibodies were described in (*Dubois et al., 2017*). Peptides DKGKSVQYAKQVEIE and FSQVRKQAYKKQTQP from the C-terminus of PgmL1 and PgmL5a, respectively, were used for rabbit immunization to yield α-PgmL1 and α-PgmL5a antibodies (Eurogentec). Polyclonal antibodies were purified by antigen affinity purification. A commercial α-Flag monoclonal antibody (monoclonal anti-Flag M2 antibody, Sigma Aldrich) was used for the detection of 3X Flag fusion proteins. The specificity of α-PgmL1 and α-PgmL5a antibodies was validated by immunofluorescence using *PGML1* and *PGML5a and b*-silenced cells respectively, whereas the specificity of the α-Flag antibody was validated using non-injected control cells (*Figure 3—figure supplement 1*).

Immunofluorescence and western blot analysis were performed as described (*Dubois et al., 2017*), with these modifications. Autogamous cells from middle-scale cultures were washed with Dryl's buffer (2 mM sodium citrate, 1 mM $NaH_2PO_4$, 1 mM $Na_2HPO_4$, 1 mM $CaCl_2$), extracted with ice-cold PHEM (60 mM PIPES, 25 mM Hepes, 10 mM EGTA, 2 mM $MgCl_2$ pH 6.9)+1% Triton during 4 min, fixed for 15 min in PHEM +2% formaldehyde. Cells were further washed three times in TBST (10 mM Tris pH 7.4, 0.15 M NaCl, 0.1% Tween20)+3% BSA. The Triton-mediated pre-extraction step was found to increase the detection signal and lower the background level for all the antibodies we used. Antibody incubation was done in TBST +3% BSA for 2 hr at room temperature using either α-Pgm 2659-GP (1:500), α-PgmL1 (1:800), α-PgmL5a (1:800) or α-Flag (1:500) antibodies. Primary antibodies were detected with (Alexafluor 488)-conjugated goat anti-guinea pig, anti-rabbit or anti-mouse IgG (1:500, ThermoFisher Scientific) and DNA was counterstained with 0.5 µg/ml DAPI (Sigma). Epifluorescence microscopy was performed as described (*Dubois et al., 2017*). The size of developing MACs (in $\mu m^2$) was measured at the maximal area section and quantification of the Pgm signal was performed using the ImageJ software (https://imagej.nih.gov/). The mean Pgm fluorescence intensity corresponds to the mean fluorescence intensity (per surface unit) in a developing MAC minus the mean extracellular background fluorescence intensity on the slide for each RNAi condition. For each experiment, normalization was performed using the mean value obtained for the corresponding control dataset. Boxplots were drawn using BoxPlotR (http://boxplot.bio.ed.ac.uk/). Mann-Whitney-Wilcoxon statistical tests (*Marx et al., 2016*) (https://ccb-compute2.cs.uni-saarland.de/wtest/) were performed to compare the datasets obtained under different conditions.

For the protein extraction from *Paramecium* cells and western blot analysis, 3 to $6 \times 10^5$ autogamous cells were collected by centrifugation at T5-T10 and washed with Dryl's buffer before transfer to liquid nitrogen. Frozen concentrated cells were directly lysed following addition of an equal volume of boiling 10% SDS containing 1x Protease Inhibitor Cocktail Set 1 (Merck Chemicals) and incubated at 100℃ for 3 min. SDS-PAGE and western blotting with α-Pgm 2659-GP (1:500) and α-alpha Tubulin TEU435 (1:1000) were performed as described (*Dubois et al., 2017*). The signal was visualized with the ChemiDoc Touch Imaging System (Bio-rad) and densitometric analyses were performed using Image Lab software (Bio-rad).

## Protein expression in insect cells and co-precipitation assays

For MBP-Pgm or MBP expression, plasmids pVL1392-MBP-PGM and pVL1392-MBP (*Marmignon et al., 2014*) were transfected individually into High Five cells together with the BD BaculoGold Linearized Baculovirus DNA (BD Biosciences) to produce recombinant baculoviruses (*Dubois et al., 2017*).

Synthetic *PGML* genes adapted to the universal genetic code (Eurofins Genomics, *Supplementary file 5*) were cloned into the pFastBAC vector (ThermoFisher Scientific) and fused at their 5' end to a nucleotide sequence encoding the HA tag. Production of recombinant baculoviruses and expression of HA-PgmL fusions were performed using the BAC-to-BAC baculovirus expression system (ThermoFisher Scientific).

To co-express each HA-PgmL fusion with MBP-Pgm (or the MBP control), High Five cells were co-infected with the appropriate recombinant baculoviruses. Cell lysis, preparation of soluble protein extracts, co-precipitation on amylose beads and detection of HA-tagged PgmLs on western blots using HA-7 monoclonal α-HA antibodies (Sigma Aldrich) were performed as described (*Dubois et al., 2017*). Independent experiments showed that the HA epitope does not interact nonspecifically with MBP-PGM.

## High throughput sequencing and analysis of IES retention

For each *PGML* KD, total genomic DNA was extracted from vegetative parental cells or nuclear preparations enriched in developing MACs from the same cultures at late autogamy stages (following 4 days of starvation). DNA was sequenced at a 76 to 160X coverage by a paired-end strategy using Illumina HiSeq (paired-end read length:~2×100 nt) or NextSeq (paired-end read length:~2×150 nt) sequencers. Sequencing reads were mapped against the MAC or MAC + IES reference genomes of *P. tetraurelia* 51 (*Arnaiz et al., 2012*). IRS correspond to the mean of the two boundary scores of a given IES calculated using the MIRET module of the ParTIES package (*Denby Wilkes et al., 2016*). Because variable amounts of DNA from old MAC fragments are present in the samples, the retention scores calculated in each experiment cannot be considered as absolute measurements of IES retention in the new MAC. For each IES, boundary scores were individually compared to those obtained in a control autogamy experiment performed in standard *K. pneumoniae* medium, as described in (*Gruchota et al., 2017*) and a statistical test for the significance of each boundary score was performed using the ParTIES package. This allowed us to define two groups of IESs: a set of significantly retained IESs and a set of 'non-retained' IESs (*i.e.* excised) that did not pass the statistical test.

Excision errors were analyzed using the MILORD module of ParTIES, with the mapping performed on the MAC + IES reference. Each error was counted as 1, independently of the number of corresponding reads. An estimate number of de novo excision errors in the new MAC was calculated by removing the errors already found in the MAC of parental vegetative cells from those that were detected in total genomic DNA from autogamous cells (*Supplementary file 9*). De novo error counts were normalized relative to the total number of sequence reads for each sample.

## Data availability

All DNA-seq datasets (*Supplementary file 8*) generated in this study were deposited in the European Nucleotide Archive under the Project Accession PRJEB24171. Reference genomes and IESs are available through ParameciumDB (https://paramecium.i2bc.paris-saclay.fr).

## Acknowledgments

This work has benefited from the facilities and expertise of the high throughput sequencing core facility of I2BC (Centre de Recherche de Gif - http://www.i2bc-saclay.fr/). We thank Sylvain Bouvard, Franck Mayeux and Victor Plet for performing preliminary experiments during their master internship, Cindy Mathon and Pascaline Tirand for excellent technical assistance and Arthur Abello, Marc Guérineau, Sandra Duharcourt, Laurent Duret, Eric Meyer, Nelly Morellet, Anne-Marie Tassin and all members of the Duharcourt, Meyer and Tassin laboratories for stimulating discussions. The project was carried out in the framework of the CNRS GDRI 'Paramecium Genome Dynamics and Evolution'.

## Additional information

### Funding

| Funder | Grant reference number | Author |
|---|---|---|
| Agence Nationale de la Recherche | ANR-10-BLAN-1603 | Mireille Bétermier |
| Fondation ARC pour la Recherche sur le Cancer | #PJA20151203521 | Julien Bischerour |
| European Research Council | 260358 | Mariusz Nowacki |
| Schweizerischer Nationalfonds zur Förderung der Wissenschaftlichen Forschung | 31003A_146257 | Mariusz Nowacki |
| Centre National de la Recherche Scientifique | Intramural CNRS funding | Mireille Bétermier |
| Agence Nationale de la Recherche | ANR-12-BSV6-0017 | Linda Sperling |

Schweizerischer Nationalfonds    31003A_166407                    Mariusz Nowacki
zur Förderung der Wis-
senschaftlichen Forschung

The funders had no role in study design, data collection and interpretation, or the
decision to submit the work for publication.

## Author contributions

Julien Bischerour, Conceptualization, Funding acquisition, Investigation, Visualization, Writing—origi-
nal draft, Writing—review and editing, Designed, Performed and analyzed wet-lab experiments; Sim-
ran Bhullar, Investigation, Writing—review and editing, Designed, Performed and analyzed wet-lab
experiments; Cyril Denby Wilkes, Software, Formal analysis, Visualization, Analyzed all DNA
sequencing data; Vinciane Régnier, Investigation, Visualization, Methodology, Writing—review and
editing, Designed, Performed and analyzed wet-lab experiments; Nathalie Mathy, Investigation;
Emeline Dubois, Validation, Writing—review and editing; Aditi Singh, Validation; Estienne Swart,
Investigation, Writing—review and editing, Identified PGML genes in Paramecium species; Olivier
Arnaiz, Data curation, Identified PGML genes in Paramecium species; Linda Sperling, Data curation,
Funding acquisition, Writing—review and editing, Identified PGML genes in Paramecium species;
Mariusz Nowacki, Supervision, Funding acquisition, Directed the experiments performed in Bern;
Mireille Bétermier, Conceptualization, Funding acquisition, Investigation, Visualization, Writing—
original draft, Project administration, Writing—review and editing

## Author ORCIDs

Julien Bischerour ⓘ https://orcid.org/0000-0002-5254-3395
Olivier Arnaiz ⓘ https://orcid.org/0000-0002-9626-1015
Mireille Bétermier ⓘ http://orcid.org/0000-0002-5407-6292

## Decision letter and Author response

Decision letter https://doi.org/10.7554/eLife.37927.043
Author response https://doi.org/10.7554/eLife.37927.044

# Additional files

## Supplementary files

• Supplementary file 1. MUSCLE alignment of the transposase core domains of ciliate domesticated
PB transposases and other PB transposases
DOI: https://doi.org/10.7554/eLife.37927.024

• Supplementary file 2. Table of Pgm and PgmL proteins encoded by published *Paramecium*
genomes and their ParameciumDB accession numbers
DOI: https://doi.org/10.7554/eLife.37927.025

• Supplementary file 3. Sequences of the cysteine-rich domains used for the alignment shown in *Fig-
ure 1—figure supplement 1*
DOI: https://doi.org/10.7554/eLife.37927.026

• Supplementary file 4. Sequences of the transposase core domains used for the alignement shown
in *Supplementary file 1*.
DOI: https://doi.org/10.7554/eLife.37927.027

• Supplementary file 5. Sequence of the synthetic *PGML* genes used for protein production in insect
cells
DOI: https://doi.org/10.7554/eLife.37927.028

• Supplementary file 6. Analysis of post-autogamous progeny in small-scale PGML knockdowns,
DOI: https://doi.org/10.7554/eLife.37927.029

• Supplementary file 7. Analysis of post-autogamous progeny in middle- and large-scale *PGML*
knockdowns
DOI: https://doi.org/10.7554/eLife.37927.030

• Supplementary file 8. DNA-seq datasets from ENA project PRJEB24171 (this study)

DOI: https://doi.org/10.7554/eLife.37927.031

• Supplementary file 9. Analysis of IES excision reads in *PGM* and *PGML* knockdowns

DOI: https://doi.org/10.7554/eLife.37927.032

• Transparent reporting form

DOI: https://doi.org/10.7554/eLife.37927.033

## Data availability

All DNA-seq datasets generated in this study were deposited in the European Nucleotide Archive under the Project Accession PRJEB24171. Reference genomes and IESs are available through ParameciumDB (http://paramecium.i2bc.paris-saclay.fr).

The following dataset was generated:

| Author(s) | Year | Dataset title | Dataset URL | Database, license, and accessibility information |
|---|---|---|---|---|
| Bischerour J, Bhullar S, Denby Wilkes C, Régnier V, Mathy N, Dubois E, Singh A, Swart E, Arnaiz O, Sperling L, Nowacki M, Bétermier M | 2018 | DNA-seq of PGMLs knocked down cells | http://www.ebi.ac.uk/ena/data/view/PRJEB24171 | Publicly available at the European Nucleotide Archive (accession no: PRJEB24171) |

The following previously published datasets were used:

| Author(s) | Year | Dataset title | Dataset URL | Database, license, and accessibility information |
|---|---|---|---|---|
| Arnaiz O, Mathy N, Baudry C, Malinsky S, Aury JM, Denby Wilkes C, Garnier O, Labadie K, Lauderdale BE, Le Mouël A, Marmignon A, Nowacki M, Poulain J, Prajer M, Wincker P, Meyer E, Duharcourt S, Duret L, Bétermier M, Sperling L | 2012 | DNA-seq of PGM knocked down cells | http://www.ebi.ac.uk/ena/data/view/ERA137444 | Publicly available at the European Nucleotide Archive (accession no: ERA137444) |
| Arnaiz O, Mathy N, Baudry C, Malinsky S, Aury JM, Denby Wilkes C, Garnier O, Labadie K, Lauderdale BE, Le Mouël A, Marmignon A, Nowacki M, Poulain J, Prajer M, Wincker P, Meyer E, Duharcourt S, Duret L, Bétermier M, Sperling L | 2012 | DNA-seq strain 51MAC | http://www.ebi.ac.uk/ena/data/view/ERA137420 | Publicly available at the European Nucleotide Archive (accession no: ERA137420) |

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
