## [Decision Letter]

Thank you for submitting your article "Six domesticated PiggyBac transposases together carry out programmed DNA elimination in *Paramecium*" for consideration by *eLife*. Your article has been reviewed by two peer reviewers, including Bernard de Massy as the Reviewing Editor and Reviewer #1, and the evaluation has been overseen by Jessica Tyler as the Senior Editor.

The reviewers have discussed the reviews with one another and the Reviewing Editor has drafted this decision to help you prepare a revised submission.

Summary:

The authors' group previously reported that the domesticated PiggyBac transpose Pgm is required for the programmed genome rearrangement (IES elimination) in *Paramecium*. In this manuscript, the authors identified and characterised additional five groups of domesticated PiggyBac transposes, PgmL1, 2, 3, 4 and 5, in *Paramecium*. All these PgmLs most likely are catalytically inactive because they lack the endonuclease "DDD" residues. The authors showed that all PgmLs can be co-precipitated with Pgm when they were co-expressed in insect cells. All the PgmL proteins localised to the new macronuclei, where programmed genome rearrangement occurs. Individual RNAi knockdown of each PgmL (or PgmL group) inhibited: 1) the production of functional new macronucleus; 2) partially (in PgmL1 KD) or severely (in PgmL2, 3, 4 or 5 KD) the accumulation of Pgm in the new macronucleus; and 3) severely (but not completely) the eliminations of IESs. The residual IES elimination events in the absence of PgmL1 or PgmL3 were accompanied with high frequency of errors in which excision boundaries tend to shift by ~10-11 bp internal to IESs. The authors concluded that PgmLs act together with Pgm to carry out precise IES excisions and that PgmL1 and 3 are important for proper positioning of the complex.

Essential revisions:

Overall this is an interesting study providing new insight into molecular components involved in IES excision in *Paramecium*. The understanding of the mechanistic contribution of the PgmL proteins is still limited but these results are definitely suitable for publication providing several features of the phenotypes are clarified.

1) The conclusion on chromatin association on fixed cells is not convincing as impact of Triton can vary greatly between proteins. In addition, the authors conclude an exacerbation with Triton (from 50-60% to 85% decrease): what is the statistical validation for this difference?

Well established protocols exist to purify soluble chromatin and chromatin bound fractions (with inclusion of control proteins) and this should be used to establish the phenotype in PgmL depleted cells.

2) 3B: The interaction analysis requires additional validations:

Indicate that WB was done with anti-HA; Add MW markers.

Other issues with this experiment: The use of 500mM salt does not always avoid interaction with DNA as some proteins can bind DNA at high salt, so this is not a convincing argument; The level of purification should be presented, because any contaminant co-purifying with MBP-Pgm could be responsible for interaction with PgmL. How do the authors exclude this possibility? Also control should be included with use of another HA tagged protein, as HA could interact with Pgm.

3) Figure 5. A PgmKD should be shown (data is in Figure 5—figure supplement 3) to compare with PgmLKD and to evaluate what is the background signal.

The authors suggest that excision events observed in PgmL1 and 3KD are due to partial depletion, and support this conclusion based on the partial KD of PgmL2 which is convincing. The authors also conclude that PgmL1 and 3 have a specific role based on the difference of excision events observed in these KD and the partial PgmL2KD. This conclusion is based on the comparison of size distribution and is not convincing. Although the distributions do appear in part different, it is not possible to tell what is reproducible and statistically significant:

The authors should confirm the specificity of PgmL1 and 3 by analyzing the type of errors observed in partial KD of PgmL2: it is expected that the use of 10bp shifted TA is not observed. This point is essential to validate the model presented in Figure 7.

4) The model figure (Figure 7) is misleading because i) presence of the all PgmLs and Pgm in a same complex has not been demonstrated; ii) it was previously shown that Pgm can form homo-oligomer; iii) all of the PgmLs can directly bind to Pgm; iv) interactions among PgmLs have not been analysed.

The middle section (PgmL1 and 3KD) is also confusing: the authors aim to summarize the consequences of PgmL1 and 3 KD, but since they have distinct properties, they present only the complex in the absence of PgmL1 but arrows refer to 1 and 3.

5) Because PgmL1 and 3KD have lower IES excision frequency, it is not clear what is the actual error rate (after normalization for excision rate)?

6) Insert the information about the general consensus for IES ends (subsection “PgmL1- or PgmL3a&b-depleted cells are prone to IES excision errors”, last paragraph).

7) Figure 6E: it would be interesting to have a plot of the fraction of IE with errors among excised IES.

8) Discussion: To conclude of a defect in nuclear import is an overstatement and far to be demonstrate by the data; Discussion about specific roles of PgmL1 and 3 (subsection “IES excision in *Paramecium* is carried out by multiple partners including Pgm and five novel domesticated PB transposases”, last paragraph) not so convincing because KD are partial.

Other comments

9) Figure 1A: The drawing of the first line (Pfam domain) is not clear: Where is DDE_Tnp_1_7?; The RnaseH fold is in orange? The DDE_Tnp_1-like zinc ribbon is in grey?

What do black vertical lines represent, presumably the catalytic D? Why only two lines in PB? Why none in Pgm?

10) Figure 1B: What does "?" mean?

11) Please explain in the main text what are T0, T5, T11. Impossible to understand for non-aficionados.

12) Transgene expression the GFP-fusion PGML upon injection into MAC of vegetative cells. Could the authors briefly explain what happens to these transgenes, if they are integrated? In which genome?

13) Figure 3A: Why silencing of ac has less effect than silencing a alone?

Statistical test for% of progeny.

14) 5A: Legend should indicate that grey bars represent IES that are not significantly excised (if I understood properly). Can you clarify the statement:

"Because variable amounts of DNA from old MAC fragments are present in the samples, the retention scores calculated in each experiment cannot be considered as absolute measurements of IES retention in the new MAC.": What is the range of variation, has it been quantified, if so could this be used to normalize retention score.

This is confusing and no explanation is provided as to why distribution of scores differs between Pgml2 and 4 for instance.

Also, an explanation is needed to understand that at the same retention score some as statistically significantly not retained, some are not (grey vs. magenta). Is this due to number of reads for different IES? It seems that the different IES are plotted by bins of retention score windows? Please clarify?

Add KD next to genes.

15) Figure 5C: significantly longer excised IES: which statistical test?

16) Figure 5D: Absence of 46-47 in PgmL: not very clear (in particular for Pgml3 KD): zoom out on the graph. Why is it important? What is the implication?

17) Figure 6E is a plot of partial internal errors not only of the 10 to 11bp shift as written in the text (subsection “PgmL1- or PgmL3a&b-depleted cells are prone to IES excision errors”, last paragraph).

18) Update Morellet 2018 reference.

19) Figure 3—figure supplement S1: What does the cross mean?

20) Subsection “*PGML* KDs have a genome-wide impact on IES elimination”, last paragraph: Cannot find progeny survival data in Figure 1—figure supplement 3?

21) Why are the correlation coefficients of PgmL2KD with other KD and with the partial KD so low?

22) Some of the supplementary figures are not mentioned in the main text, please revise or remove.

---

## [Author Response]

Essential revisions:[…] 1) The conclusion on chromatin association on fixed cells is not convincing as impact of Triton can vary greatly between proteins. In addition, the authors conclude an exacerbation with Triton (from 50-60% to 85% decrease): what is the statistical validation for this difference?

We first would like to emphasize that all experiments address the nuclear localization of the same protein (Pgm) under different RNAi conditions. We have now included statistical analyses (Mann-Whitney-Wilcoxon test) of the Pgm immunofluorescence nuclear signals displayed in Figure 4C and 4D. We also present in Figure 4—figure supplement 2D a statistical comparison of the mean normalized values obtained with and without Triton pre-extraction. This analysis confirms that a statistically significant decrease of Pgm nuclear signal is observed following Triton pre-extraction in *PGML2/3/4/5* RNAi relative to conditions omitting the pre-extraction step, while Triton treatment has no significant effect on Pgm nuclear localization in *PGML1* RNAi.

Well established protocols exist to purify soluble chromatin and chromatin bound fractions (with inclusion of control proteins) and this should be used to establish the phenotype in PgmL depleted cells.

We agree with the reviewers that definite proof of a defect in Pgm association with chromatin in *PGML* RNAi requires additional biochemical evidence.

However, we must stress here that there are no well-established protocols in *Paramecium* to extract purified soluble chromatin and chromatin-bound fractions. Neither are there, to our knowledge, available antibodies that would recognize a control *Paramecium* chromatin-bound protein. Ciliate and mammalian proteins are generally poorly conserved (Author response image 1), which renders the use of commercial antibodies only rarely feasible.

**Author response image 1. respfig1:** *Paramecium* proteins are highly divergent relative to their mammalian orthologs. Percent identity between orthologs was calculated using the InParanoid tool from Cildb (Arnaiz et al., 2009, Database 2000 bap022; Arnaiz et al., 2014, Cilia 3:9; http://cildb.cgm.cnrs-gif.fr/). Dotted lines indicate the median of each distribution. Overall, *Paramecium* proteins share only 35% identity (median) with mammalian proteins, while mouse and human proteins share 95% identity with each other.

We nonetheless prepared nuclear extracts from control and PgmL-depleted autogamous cells and tried two procedures to extract endogenous Pgm. We performed a differential salt fractionation (Hermann et al., 2017, Bio Protoc 7 pii: e2175), as well as an extraction using a modified Wuarin-Schibler buffer (MWS; Gagnon et al., 2014, Nat Protoc 9: 2045-2060). As shown for control cells in Author response image 2, Pgm remained in the insoluble fraction under all extraction conditions tested, whereas histone H3 started to be solubilized at 300mM NaCl in the salt fractionation protocol (lane 3). This suggests that Pgm is insoluble under these conditions, consistent with our independent observation that proper Pgm detection in total extracts or in purified nuclei requires three-minute boiling in 5% SDS, prior to denaturation in Laemmli buffer. We observed that Pgm behaved similarly in PgmL-depleted cells. For this reason, we cannot – for the moment – address biochemically the chromatin-associated status of Pgm.

Accordingly, we have removed our initial statement that Pgm is more loosely bound to chromatin in the absence of PgmL proteins from the Results section and only refer to its nuclear localisation.

**Author response image 2. respfig2:** Attempts to extract endogenous Pgm from chromatin in control cells. *Paramecium* nuclei were purified by low-speed centrifugation of autogamous cell lysates, as described (Arnaiz et al., 2012). Immunoblotting was performed using α-Pgm 2659-GP antibodies for Pgm detection (Dubois et al., 2017) and commercial antibodies for the detection of histone H3 (Merck Millipore # 07-690).

2) 3B: The interaction analysis requires additional validations:Indicate that WB was done with anti-HA; Add MW markers.

The use of anti-HA antibodies (α-HA) has been mentioned in the figure legend and indicated in Figure 3B itself. The full-size blot with all lanes and molecular weight markers has been included in Figure 3—figure supplement 3B.

Other issues with this experiment: The use of 500mM salt does not always avoid interaction with DNA as some proteins can bind DNA at high salt, so this is not a convincing argument;

We can rule out that Pgm binds DNA at 500 mM NaCl. Indeed, we performed a control DRaCALA DNA binding assay (Differential Radial Capillary Action of Ligand Assay, see Donaldson et al., 2012), which we have included in Figure 3—figure supplement 3A. While Pgm clearly binds DNA at low NaCl concentration (50 or 100 mM), no DNA binding was detected at 250 mM NaCl and above. We are, therefore, confident that co-purification of Pgm and PgmLs cannot be explained by simultaneous binding of both protein to the same DNA molecule. We have included these clarifications in the Results section.

The level of purification should be presented, because any contaminant co-purifying with MBP-Pgm could be responsible for interaction with PgmL. How do the authors exclude this possibility?

We are aware that a 100% purity level has not been reached in the co-precipitates and agree with the reviewers that the presence of a contaminating partner cannot formally be excluded. We would like to point out, however, that all assays were performed in heterologous insect cell extracts containing no other *Paramecium* protein than Pgm and PgmLs. We consider it unlikely that an insect contaminating protein strongly bridges Pgm and PgmL.

Also control should be included with use of another HA tagged protein, as HA could interact with Pgm.

The requested negative control has been provided for review, but has not been included here since the experiment was designed for another study.

3) Figure 5. A PgmKD should be shown (data is in Figure 5—figure supplement 3) to compare with PgmLKD and to evaluate what is the background signal.

The distribution obtained in a previously published *PGM* KD (Arnaiz et al., 2012) has been included in Figure 5A. This indeed corresponds to the data shown in Figure 5—figure supplement 3.

The authors suggest that excision events observed in PgmL1 and 3KD are due to partial depletion, and support this conclusion based on the partial KD of PgmL2 which is convincing.

There might have been some misunderstanding here. Residual excision activity can indeed be observed in a partial *PGML2* KD. We believe, however, that partial depletion of PgmL1 or PgmL3 *cannot* be the sole explanation for the error-prone IES excision events observed in *PGML1* or *PGML3* KDs. Indeed, over-representation of one particular type of errors (i.e. partial internal) is clearly specific for these two KDs and is not observed in partial *PGML2* KDs. This analysis is displayed in Figure 6—figure supplement 2.

The authors also conclude that PgmL1 and 3 have a specific role based on the difference of excision events observed in these KD and the partial PgmL2KD. This conclusion is based on the comparison of size distribution and is not convincing. Although the distributions do appear in part different, it is not possible to tell what is reproducible and statistically significant:The authors should confirm the specificity of PgmL1 and 3 by analyzing the type of errors observed in partial KD of PgmL2: it is expected that the use of 10bp shifted TA is not observed. This point is essential to validate the model presented in Figure 7.

The conclusion that PgmL1 and 3 have a specific role in IES excision is actually based on two observations:

i) In *PGML1* or *PGML3* KDs, the population of efficiently excised IESs indeed shows a significantly different size distribution relative to strongly retained IESs, with an enrichment in >77-bp IESs. A statistical comparison of the size distributions of the two IES populations is presented in Figure 5C (Mann-Whitney-Wilcoxon test).

ii) Figure 6 and Figure 6—figure supplement 2 show that partial internal excision errors are specifically over-represented among all excision errors in *PGML1* or *PGML3* KDs. To address the reviewers’ concern, we have now included an additional panel (panel B) in Figure 6—figure supplement 2 to show that no specific 10-bp shift is observed in partial *PGML2* KDs.

4) The model figure (Figure 7) is misleading because i) presence of the all PgmLs and Pgm in a same complex has not been demonstrated; ii) it was previously shown that Pgm can form homo-oligomer; iii) all of the PgmLs can directly bind to Pgm; iv) interactions among PgmLs have not been analysed.The middle section (PgmL1 and 3KD) is also confusing: the authors aim to summarize the consequences of PgmL1 and 3 KD, but since they have distinct properties, they present only the complex in the absence of PgmL1 but arrows refer to 1 and 3.

The reviewers are right. We are aware that more work is needed to provide deeper insight into the organization and stoichiometry of PgmL subunits in the Pgm-associated complex. We have prepared a simplified version of Figure 7, in which no assumption is made with regard to the exact number of PgmLs present in the complex. In line with previously published data (Dubois et al., 2017) and with known properties of the canonical PiggyBac transposase from *Trichoplusia ni* (Jin et al., 2017), we have drawn Pgm as a dimer positioned at each TA cleavage site. The revised model has been focused on the control of DNA cleavage at IES ends. As stated above, we agree with other reviewers’ comments (see reply to point 1) that speculating on the role of PgmL in stabilizing Pgm association with chromatin is premature at this stage. The first paragraph of the Discussion has been modified accordingly.

5) Because PgmL1 and 3KD have lower IES excision frequency, it is not clear what is the actual error rate (after normalization for excision rate)?

Due to experimental limitations, it is not possible to calculate IES excision rates precisely. Indeed, the DNA used for high throughput sequencing is prepared from nuclear preparations enriched in developing MACs, but fragments of the old MAC always contaminate these preparations. Thus, in our sequencing datasets, IES^-^ reads do not exclusively represent de novo excision events that took place in the developing MACs. Furthermore, because the contamination is variable from one sample to the other, no simple correction can be applied to our data to accurately remove the contribution of the old MAC.

6) Insert the information about the general consensus for IES ends (subsection “PgmL1- or PgmL3a&b-depleted cells are prone to IES excision errors”, last paragraph).

Done.

7) Figure 6E: it would be interesting to have a plot of the fraction of IE with errors among excised IES.

This information was available in our initial Figure 6—figure supplement 1. To make it more conspicuous, we have prepared a new version of this figure, which now includes the requested plot as an additional panel (panel B). We explicitly refer to the data in the main text (subsection “PgmL1- or PgmL3a&b-depleted cells are prone to IES excision errors”).

8) Discussion: To conclude of a defect in nuclear import is an overstatement and far to be demonstrate by the data; Discussion about specific roles of PgmL1 and 3 (subsection “IES excision in Paramecium is carried out by multiple partners including Pgm and five novel domesticated PB transposases”, last paragraph) not so convincing because KD are partial.

We agree that not enough evidence can be provided for a defect in Pgm nuclear import. We have removed this suggestion from the Results and Discussion sections. We have also removed all considerations about the protein import/export balance from the model of Figure 7, which now puts more emphasis on the biochemical properties of the Pgm-associated complex.

As stated above (reply to Point 3), our work clearly shows that *PGML1* or *PGML3* KDs induce specific phenotypes that are not observed in partial *PGML2* KDs and, therefore, cannot simply be explained by partial depletion. We have put more emphasis on this point in the Discussion.

Other comments9) Figure 1A: The drawing of the first line (Pfam domain) is not clear: Where is DDE_Tnp_1_7?; The RnaseH fold is in orange? The DDE_Tnp_1-like zinc ribbon is in grey?What do black vertical lines represent, presumably the catalytic D? Why only two lines in PB? Why none in Pgm?

The Pfam domain DDE_Tnp_1_7 is shown as a bipartite orange domain, with the RNaseH fold corresponding to its right part. The DDE_Tnp_1-like zinc ribbon is in grey. These explanations are now provided in the figure legend. We completed the figure by adding vertical bars to represent all conserved D residues.

10) Figure 1B: What does "?" mean?

The question mark indicates that the expected α4 helix of the RNaseH fold domain could not be predicted using the PSIPRED secondary structure program. This is now specified in the legend.

11) Please explain in the main text what are T0, T5, T11. Impossible to understand for non-aficionados.

The explanation was mentioned in the Materials and methods section. We moved it to the main text and completed the legend of Figure 2 by specifying that all time-points are in hours. We also explained in the Introduction that the MAC “is fragmented and destroyed at each sexual cycle” before a new MAC develops from a copy of the zygotic nucleus.

12) Transgene expression the GFP-fusion PGML upon injection into MAC of vegetative cells. Could the authors briefly explain what happens to these transgenes, if they are integrated? In which genome?

We have added the following explanation to the legend of Figure 2—figure supplement 2:

“Following microinjection into the MAC of vegetative cells, transgenes are capped by addition of telomeric repeats at their ends (concatemers may form prior to telomere addition): they may then integrate into the somatic genome (through a mechanism that remains to be studied) or be maintained throughout vegetative growth as autonomously replicating mini-chromosomes (Gilley et al. 1988; Bourgain and Katinka 1991; Katinka and Bourgain 1992). […] Transgenes persist in old MAC fragments throughout autogamy and continue to be expressed if controlled by proper transcription signals, but are lost in the next sexual generation, after complete destruction of the old MAC”.

13) Figure 3A: Why silencing of ac has less effect than silencing a alone?Statistical test for% of progeny.

Within the *PGML3* group, *PGML3a* shows by far the highest expression level (Figure 2A), which is the reason which silencing *a* leads to a strong – although partial – phenotype. *PGML3c* has a low expression level and its specific contribution to total PgmL3 amounts is probably very small. In addition, the double feeding procedure that we used to silence *a* and *c* consists in feeding *Paramecium* cells with a 1:1 mix of induced bacteria producing dsRNA homologous to each gene. In practice, this leads to a 2-fold dilution of *PGML3a*-inducing bacteria, which might explain why silencing *3ac* appears to have less effect than silencing *3a* alone (p=0.053 in a two-sample t-test). The purpose of Figure 3A is to show that silencing of the two most expressed genes within each *PGML* group is required to completely abolish progeny survival. For the sake of clarity, we have not systematically mentioned the results of the statistical analysis of% progeny with functional new MAC.

14) 5A: Legend should indicate that grey bars represent IES that are not significantly excised (if I understood properly).

We indicated in the legend that grey bars represent the distribution of all IESs over all possible retention scores ranging from 0 to 1 (by bins of 0.025). When relevant, the distribution of non-significantly retained (=efficiently excised) IESs is superimposed in magenta.

Can you clarify the statement:"Because variable amounts of DNA from old MAC fragments are present in the samples, the retention scores calculated in each experiment cannot be considered as absolute measurements of IES retention in the new MAC.": What is the range of variation, has it been quantified, if so could this be used to normalize retention score.This is confusing and no explanation is provided as to why distribution of scores differs between Pgml2 and 4 for instance.

As stated above (reply to point 5) the distributions of IES retention scores may shift along the X-axis from one experiment to the other, depending on the variable amounts of contaminating old MAC DNA present in our samples. The variability has different causes that are very difficult to control, among which the physiological state of cells (strong starvation conditions accelerate the loss of old MAC fragments) and the fragment size at the time of cell fractionation (low speed centrifugation will selectively eliminate smaller fragments). Guérin et al. (BMC Genomics 2017, 18:327) used an additional flow cytometry sorting step to enrich preparations of developing MACs from Pgm-depleted cells and obtained higher IES retention scores in their 98% pure MAC preparation than in the “non-sorted” sample shown in Figure 5A. This procedure will certainly improve future analyses, without completely eliminating the problem of quantifying the level of contamination by old MAC DNA in different preparations. This makes normalization still problematic at this stage.

Also, an explanation is needed to understand that at the same retention score some as statistically significantly not retained, some are not (grey vs. magenta). Is this due to number of reads for different IES?

As explained in the Materials and methods, our statistical analysis was performed by comparing each *PGML* KD dataset to a control obtained following autogamy in standard culture medium. For each IES, the number of reads corresponding to IES^+^ and IES^-^ molecules were compared between the two samples and a statistical test for the significance of each boundary score was performed using the published ParTIES package (Denby Wilkes et al., 2015). Given that the number of reads covering an IES differs between samples and from one IES to the other, the significance of IES retention may differ for IESs showing similar retention scores.

It seems that the different IES are plotted by bins of retention score windows? Please clarify?

This is right. The IRS bins are equal to 0.025. This is now clearly stated in the legend to Figure 5.

Add KD next to genes.

Done.

15) Figure 5C: significantly longer excised IES: which statistical test?

We performed Mann-Whitney-Wilcoxon statistical tests to compare IES size distributions. This is now stated in the figure legend.

16) Figure 5D: Absence of 46-47 in PgmL: not very clear (in particular for Pgml3 KD): zoom out on the graph. Why is it important? What is the implication?

We agree that this is a minor point. Since we have no straightforward interpretation for this observation, we have removed this statement from the Results section.

17) Figure 6E is a plot of partial internal errors not only of the 10 to 11bp shift as written in the text (subsection “PgmL1- or PgmL3a&b-depleted cells are prone to IES excision errors”, last paragraph).

This is true sensu stricto. However, as shown in Figure 6D, partial internal errors consist mostly in excision events using a 10 to 11-bp shifted TA. We changed the sentence to “Finally, we noticed that error-prone IESs, for which an internal TA (mostly shifted by 10 to 11 bp) is used in erroneous excision events, follow a different size distribution…”

18) Update Morellet 2018 reference.

Done.

19) Figure 3—figure supplement S1: What does the cross mean?

The cross indicates that blue inserts can target the two paralogs simultaneously. This has been included in the legend.

20) Subsection “PGML KDs have a genome-wide impact on IES elimination”, last paragraph: Cannot find progeny survival data in Figure 1—figure supplement 3?

Thanks for the notice: we were referring to Figure 4—figure supplement 3. Sorry for the confusion.

21) Why are the correlation coefficients of PgmL2KD with other KD and with the partial KD so low?

*PGML2* KD induces a homogeneous phenotype at the genome-wide level, with similar retention scores for all IESs. The absence of correlation indicates that variations around the mean retention score are limited and probably attributable to statistical noise.

22) Some of the supplementary figures are not mentioned in the main text, please revise or remove.

Thanks for drawing our attention to this point. Some supplementary figures were only mentioned in the Supplementary Information file. They are now all referred to in the main text.